# Adaptive coding across visual features during free-viewing and fixation conditions

Sunny Nigam [1] ✉, Russell Milton [1], Sorin Pojoga [1] & Valentin Dragoi [1,2] ✉

Theoretical studies have long proposed that adaptation allows the brain to effectively use the limited response range of sensory neurons to encode widely varying natural inputs. However, despite this influential view, experimental studies have exclusively focused on how the neural code adapts to a range of stimuli lying along a single feature axis, such as orientation or contrast. Here, we performed electrical recordings in macaque visual cortex (area V4) to reveal significant adaptive changes in the neural code of single cells and populations across multiple feature axes. Both during free viewing and passive fixation, populations of cells improved their ability to encode image features after rapid exposure to stimuli lying on orthogonal feature axes even in the absence of initial tuning to these stimuli. These results reveal a remarkable adaptive capacity of visual cortical populations to improve network computations relevant for natural viewing despite the modularity of the functional cortical architecture.

One influential view in neuroscience is that sensory cortical neurons are adapted to the statistics of natural stimuli[1–3]. According to this view, adaptation allows sensory neurons to make effective use of the limited range of neural responses to encode stimuli that vary widely in structure, such as those encountered in natural environments[4]. During visual perception, for instance, the exploration of natural scenes consists of successive visual fixations accompanied by changes in image statistics[5,6]. However, natural images, despite their complexity, have certain common statistical properties. That is, neighboring image patches are highly correlated in local attributes, such as orientation, contrast, or color, whereas distant image patches are only poorly correlated[7,8]. Therefore, successive fixations during natural viewing will often land on image patches of largely different structure. Orientation and color signals, two of the elementary features ubiquitously present in natural scenes, are considered to be orthogonal to each other as they are represented in distinct modules in primate mid-level visual cortex[9,10]. It is conceivable that during a typical visual fixation, neurons can be exposed to image patches dominated by oriented signals, while subsequent fixations could land on distant image patches where color is the primary feature (Fig. 1a, Supplementary Fig. 1). However, although successive fixations to orthogonal image features are ubiquitous during natural viewing (Fig. 1a), whether and how cross-

feature adaptation influences neuronal responses and stimulus coding remains unknown.

Previous adaptation studies have shown that rapid exposure to spatially correlated image patches induces short-term changes in the responses of visual cortical neurons and changes the tuning of individual cells and their stimulus discriminability[6,11]. At the population level, rapid adaptation was shown to reduce neuronal correlations and improve coding accuracy[12,13]. However, while these results were instrumental for our current understanding of the properties of the adaptive code, they originated from studies investigating how the neural code adapts to a relatively narrow set of stimuli along a single feature axis, such as orientation[14,15], contrast[16,17], motion[18,19] or color[20]. Equally important, previous adaptation studies have exclusively focused on experimental paradigms involving restricted viewing in which stimuli are presented during passive fixation, thereby lacking the naturalistic conditions encountered during free-viewing.

Here, we addressed two of the major limitations of previous adaptation studies by simultaneously recording the spiking activity of multiple neurons from superficial layers of visual cortical area V4 of awake macaque using chronically implanted Utah arrays (Fig. 1c, d). While monkeys freely viewed or fixated on stimuli of largely dissimilar

[1]Department of Neurobiology and Anatomy McGovern Medical School, University of Texas at Houston, Houston, TX 77030, US. [2]Department of Electrical and Computer Engineering, Rice University, Houston, TX 77005, US. ✉e-mail: sunny.nigam@uth.tmc.edu; valentin.dragoi@uth.tmc.edu

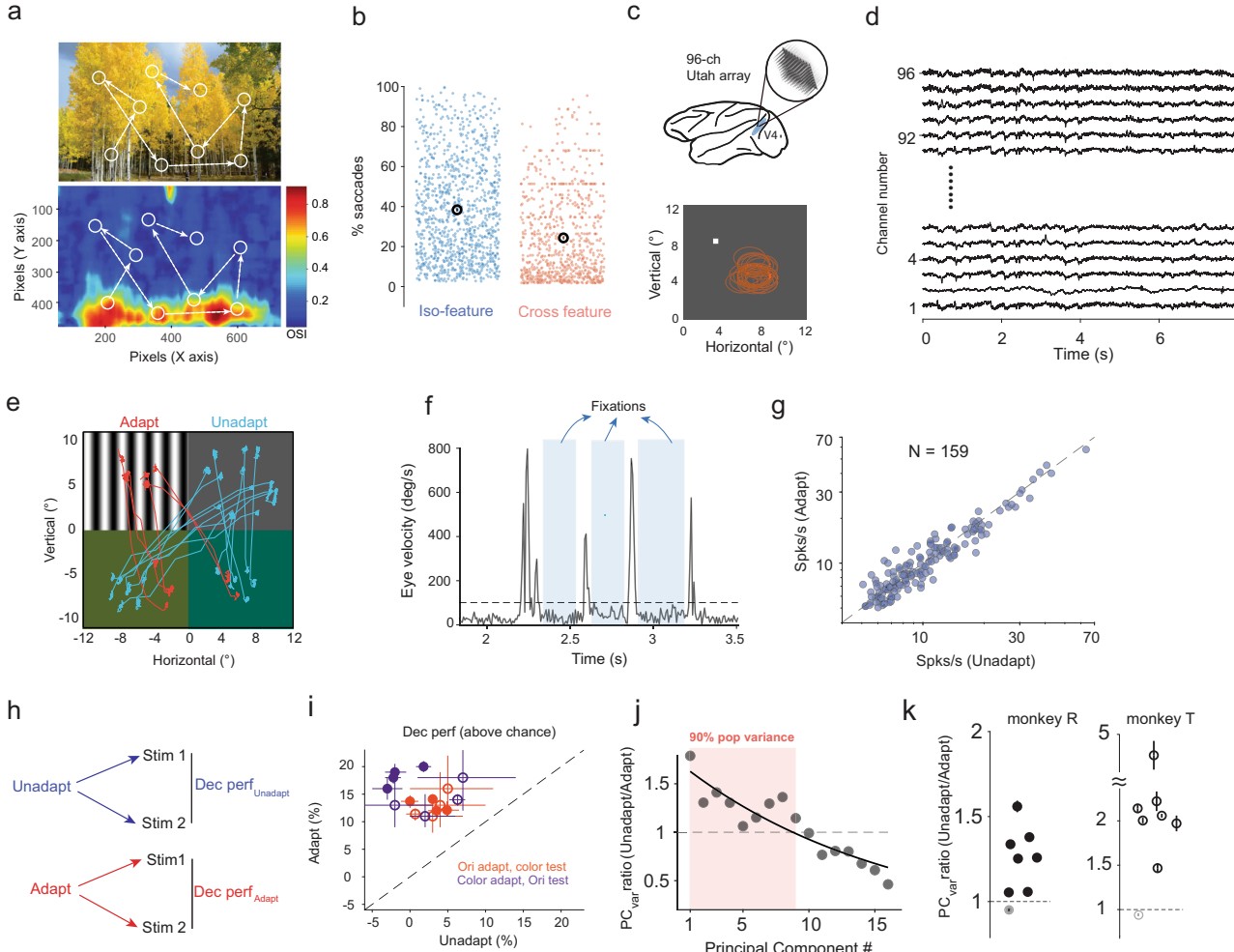

**Fig. 1 | Rapid adaptation during free-viewing improves coding accuracy in neural populations. a** Sequence of simulated saccades (arrows) with pseudo receptive field (RF) positions (circles) overlaid on natural image[49] (Olmos, A., Kingdom, F. A. A. (2004). A biologically inspired algorithm for the recovery of shading and reflectance images, Perception, 33, 1463 - 1473. http://tabby.vision.mcgill.ca/). Bottom: Heatmap of orientation selectivity index for the same image. **b** Percentage of simulated saccades connecting iso-feature (blue) and cross-feature (orange dots) patches for each natural image. Black circles and bars represent mean and s.e.m. **c** Top: Schematic of a chronically implanted 96 channel Utah array in macaque V4. Bottom: RF locations (orange circles) of single units with respect to fixation point (white dot). **d** Snippet of voltage waveforms recorded simultaneously from all 96 channels. **e** Stimulus configuration for free-viewing experiments. Red and blue traces show saccades connecting RF centers during 'adapt' and 'unadapt' conditions. **f** Example of fixation detection (shaded regions) from eye velocity trace during free-viewing. **g** Firing rates for individual neurons ($N = 159$) during una- dapted and adapted conditions ($FR_{adapt}^{mean} = 11.8$ Hz, $FR_{unadapt}^{mean} = 12.1$ Hz; two-sided

Wilcoxon signed rank test, $P > 0.05$). **h** Schematic description of decoder perfor- mance under unadapt (blue) and adapt conditions (red). **i** Decoder performance (chance level subtracted) in unadapted and adapted conditions for orientation adapter and color test stimuli (orange) and vice versa (purple). Filled and open circles represent individual sessions from monkey T and R respectively. Vertical and horizontal bars represent s.e.m. **j** Ratio of the population variance captured in trial-by-trial responses by principal components (PCs) in unadapt and adapt con- ditions for an example session (see Methods). Filled gray circles represent the ratio along each PC axis. Shaded region and solid black line represent PCs capturing 90% of the population variance and an exponential fit (for visualization purposes). **k** Mean value of PC ratio for each individual session in monkey R (solid black circles, $N = 8$ sessions) and monkey T (open black circles, $N = 8$ sessions). Horizontal bars represent s.e.m which were evaluated by performing PCA analysis multiple times ($n = 100$) on sub-sampled trials in each condition. Lighter shaded circles represent sessions where the ratio is not significantly greater than 1 (one-sided Wilcoxon signed rank test, $P > 0.05$).

structure, we examined whether successive fixations to image patches across orthogonal feature axes induce adaptive changes in the prop- erties of the neural code.

## Results

The efficient coding hypothesis[1] predicates that visual cortical responses are adapted to the statistics of natural stimuli during vision. That is, the higher the frequency of particular stimuli encountered during viewing, the more efficient the neural mechan- isms responsible for processing those stimuli. In light of this hypothesis, we examined how likely successive fixations occurring during natural viewing could land on cross-feature stimuli (con- sidering that each fixation episode would act as rapid adaptation[6]).

To this end, we built a statistical model of visual exploration of natural scenes to examine the features encountered in consecutive fixations. We focused on quantifying the distribution of visual fixa- tions on two elementary features that are ubiquitously present in natural scenes: color and orientation. Based on the distribution of saccade amplitude and direction while monkeys are freely viewing a visual display across 16 sessions (Fig. 1e and Supplementary Fig. 2, see Methods), we simulated consecutive pseudo-saccades by using a natural image battery ($n = 940$) from the McGill calibrated color image database (McGill Vision research). For each image we ran- domly assigned a fixation starting point, and then generated 300 consecutive saccades (we analyzed a total of 282,000 saccades across the images from this database). The size of image patches was

chosen based on the mean receptive field size of our V4 neurons. For every pseudo saccade connecting two patches of a natural image, we quantified the strength of orientation and color signals in each patch (see Methods). Strikingly, we found that a significant percentage of saccades (23%) involving successively fixated image patches characterized by dissimilar features (color → orientation or orientation → color) as start and end points (Fig. 1b). This analysis indicates that during the exploration of natural images, visual cortical neurons are highly likely to be exposed to successive image patches containing dissimilar visual features (orientation and color).

Given the ubiquity of exposures to largely different features during viewing natural stimuli, we designed a naturalistic task (Fig. 1e), where monkeys freely viewed four stimuli placed on different quadrants of a visual display. The top right quadrant contained a uniform gray patch (control stimulus), whereas the top left quadrant contained an adapting stimulus (achromatic grating or color patch). The bottom two quadrants contained two test stimuli separated by 22.5° in u.v. color space (when grating adapters were used; see Methods) or by 11.25° in orientation space (when color adapters were used). The receptive fields of multiple neurons recorded simultaneously were mapped at the beginning of each recording session (Fig. 1c bottom, see Methods). The adapter and test stimuli were chosen to maximize the number of responsive units in a given session. Control trials were defined as fixations on the gray patch followed by fixations on a test stimulus in either quadrant, whereas adaptation trials consisted of fixations on the adapter patch (grating or color) followed by fixations on either of the test stimuli (Fig. 1e). Fixations were detected during free viewing by thresholding eye velocity (Fig. 1f; see Methods). Only fixations whereby neurons' aggregate receptive fields were confined within the monitor borders and within a single quadrant were considered for further analysis. We analyzed the responses of 159 single units (mean firing rate >5 spks/s for both unadapted and adapted conditions) in 2 monkeys (T and R) during 9,835 fixations recorded for 21.2 hours of free viewing across 16 sessions (8 sessions from each animal).

## Improved population coding accuracy during free-viewing

We examined the extent to which the neural population can distinguish between pairs of test stimuli in control and adaptation trials during free viewing given that there was no significant difference in the mean responses of single neurons (Fig. 1g, Wilcoxon signed rank test, $P > 0.05$). To this end, we pooled neurons' responses during fixations to each test stimulus when they were preceded by the gray (control) patch or the oriented grating adapter. To examine whether cross-feature adaptation influences stimulus coding (Fig. 1h), we calculated spike counts during fixations on each test stimuli in either condition (unadapt/adapt), and then implemented a linear decoder (Linear Discriminant Analysis; see Methods). Notably, we observed an increase in decoder performance (DP) for color stimuli when cells were adapted to a different feature, i.e., orientation, and vice-versa when color was the adapter and orientation was used for the test stimuli (Fig. 1i; $\Delta DP_{adapt\_ori} = 19 \pm 2\%$; $\Delta DP_{adapt\_color} = 30 \pm 4\%$, mean ± s.e.m). These results are surprising as orientation and color are assumed to be encoded independently in visual cortex given their modular functional organization.

To understand how cross-feature adaptation leads to improved stimulus discriminability, we analyzed the structure of trial-by-trial responses of the entire neural population in the unadapted and adapted conditions (see Supplementary Fig. 3a for a pairwise correlation analysis). We hypothesized that decoder performance in the adapted condition is improved because the population response is more decorrelated, i.e., the shared variability between neurons is reduced. To examine this, we performed a principal component analysis on the responses of neurons across trials[21,22] in the unadapted condition. Next, we projected the activity of the same

neurons in the adapted condition onto the principal component axis obtained in the unadapted condition. To quantify the changes in the overall structure of shared population wide variability, we calculated the ratio of the variance explained along each principal component axis in unadapted and adapted conditions (Fig. 1j, example session 5). Finally, we calculated the mean value of the ratios for the principal components that explained 90% of the variance in the control condition. Interestingly, we found a significant reduction in explained variance (along the same PC axis) post adaptation in the vast majority of sessions in both animals (Fig. 1k, Wilcoxon signed rank test, $P < 0.01$). A similar decrease in explained variance across conditions is found even if we considered only the first PC axis (Supplementary Fig. 3b) indicating that our analysis is robust to the number of principal components chosen. Thus, in the higher dimensional space of neural responses, there is a transition from an ellipsoid to a more spherical clustering of neuronal responses post adaptation indicating an overall decorrelation of the population response, which may be the likely mechanism for the improvement in decoder accuracy.

One possible confound is the duration of fixations on the adapter preceding each test stimulus. A mean duration of fixations on the adapter preceding test₁ that is different from that of fixations preceding test₂ is equivalent to different strengths of adaptation that could conflate the interpretation of the decoder results in Fig. 1i. However, we found no significant difference between the fixation durations preceding each test stimulus ($P > 0.01$; Wilcoxon's rank-sum test; Supplementary Fig. 4a). We further analyzed other factors that could possibly account for the changes in decoder performance not directly linked to rapid adaptation. However, there were no significant differences in the distribution of fixation durations on the post-adaptation test stimuli, mean pupil size, subthreshold eye velocity of microsaccades during fixations, number and direction of microsaccades between unadapted and adapted conditions ($P > 0.05$ for all comparisons, Wilcoxon rank sum test, Supplementary Fig. 4b–e). We also observed no systematic changes in the mean firing rates of neurons associated with direction of saccades to different test stimuli (Wilcoxon signed rank test, $P > 0.1$). Additionally, the average distance between the neurons' receptive field centers and the boundary between the two test stimuli did not differ significantly between adapted and unadapted conditions ($P > 0.05$, Wilcoxon rank sum test; Supplementary Fig. 4f). Altogether, these analyses indicate that the changes in decoder performance after rapid adaptation during natural viewing cannot be attributed to confounding factors related to eye movements or our experimental design.

## Emergence of feature tuning after adaptation to the orthogonal feature

While the free viewing task allowed us to investigate cross-feature adaptation effects under naturalistic conditions, the experimental design limited us to studying stimulus coding using only one pair of test stimuli. Therefore, to examine how cross-feature adaptation impacts overall tuning properties, such as strength of tuning and stimulus preference, we performed additional experiments involving passive visual fixation. Stimulus tuning was measured using an extended set of oriented gratings and color patches while recording the visually driven spiking activity of hundreds of neurons ($n = 523$) from the superficial layers of V4 (10 sessions in monkey T; 7 sessions in monkey M). Stimuli consisted of color patches and oriented gratings used either as adapters or test stimuli. Responses to color stimuli were quantified using a set of 16 equiluminant colors ($9.35 \pm 0.01$ cd/m²) uniformly spaced in Luv color space (steps of 22.5°) with respect to a neutral gray point[20] (Fig. 2a, Supplementary table 1; see Methods), while those to oriented stimuli were examined using 16 sinusoidal gratings uniformly spanning the 0°–180° range in steps of 11.25° (same mean luminance as the color patches and gray background). Using these stimuli, we characterized the tuning preference of neurons to

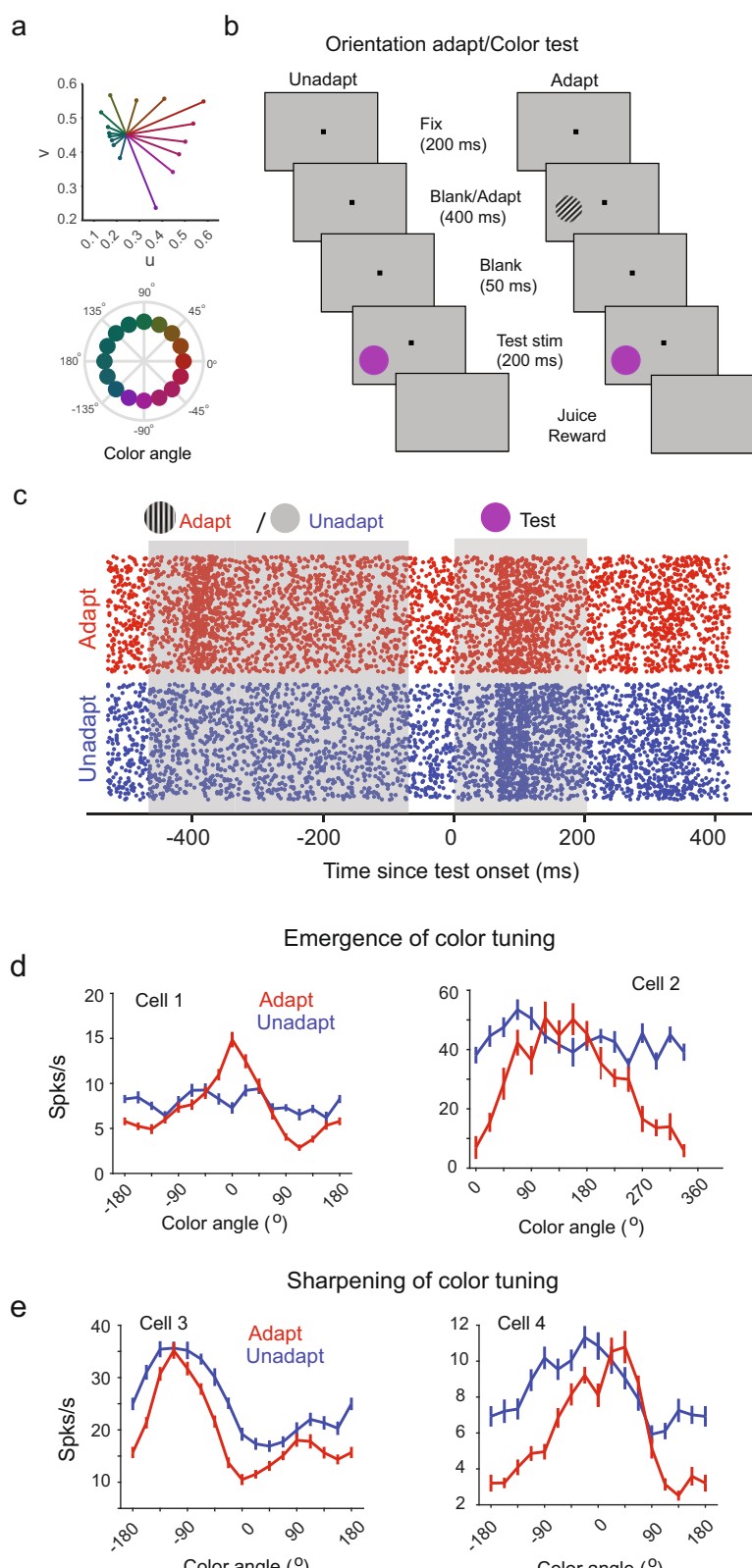

find that 63.7% of the neurons were only tuned to color, 13.2% were only tuned to orientation, 8.8% were tuned to both color and orientation, and 13.2% were not significantly tuned to either color or orientation. Each recording session consisted of randomly interleaved unadapted and adapted trials ($n = 450 \pm 30$ trials/condition). During adaptation trials, a 400-ms adapting stimulus belonging to one feature set (e.g., orientation) was followed by a 200-ms stimulus randomly

selected from the orthogonal feature set (color) with an inter-stimulus interval of 50 ms. Control (unadapted) trials did not contain an adapting stimulus; instead, a gray screen was presented for the same duration as that of the adapter (Fig. 2b, c).

Strikingly, many cells in our population either acquired color tuning or significantly sharpened it after orientation adaptation (Fig. 2d–e, see Supplementary Fig. 5 for additional examples). This is

**Fig. 2 | Emergence and enhancement of color tuning after adaptation to oriented stimuli. a** Equiluminant color stimulus set plotted in $u,v$ space. Angular plot of same color stimulus with respect to the $u,v$ coordinates of an equiluminant neutral gray background on which adapter and test stimuli are presented.
**b** Schematic description of the cross-feature adaptation task. Adapter and test stimuli are flashed across the receptive fields of recorded neurons. **c** Raster plot of an example neuron across multiple trials consisting of unadapted (blue) and adapted (red) conditions. Gray shaded regions represent the duration of the adapter (400 ms) and test stimulus (200 ms) presentation. **d** Tuning curves of

example neurons showing emergence of color tuning in color untuned neurons (cell 1, monkey T, $CSI_{adapt} = 0.26$, Rayleigh's test, pval < 0.01, $CSI_{unadapt} = 0.05$ Rayleigh's test, pval > 0.05; cell 2, monkey M, $CSI_{adapt} = 0.24$, pval < 0.01, $CSI_{unadapt} = 0.05$; pval > 0.05) after adaptation with an oriented grating. **e** Tuning curves for color stimuli in two other example neurons (cell 3, monkey T, $CSI_{adapt} = 0.21$, pval <0.01, $CSI_{unadapt} = 0.15$, pval <0.01; cell 4, monkey M, $CSI_{adapt} = 0.30$, pval <0.01, $CSI_{unadapt} = 0.13$, pval <0.01) exhibiting sharpening of color tuning after orientation adaptation. Solid lines and error bars represent the mean and s.e.m of responses to color stimuli (see also Supplementary Fig. 4).

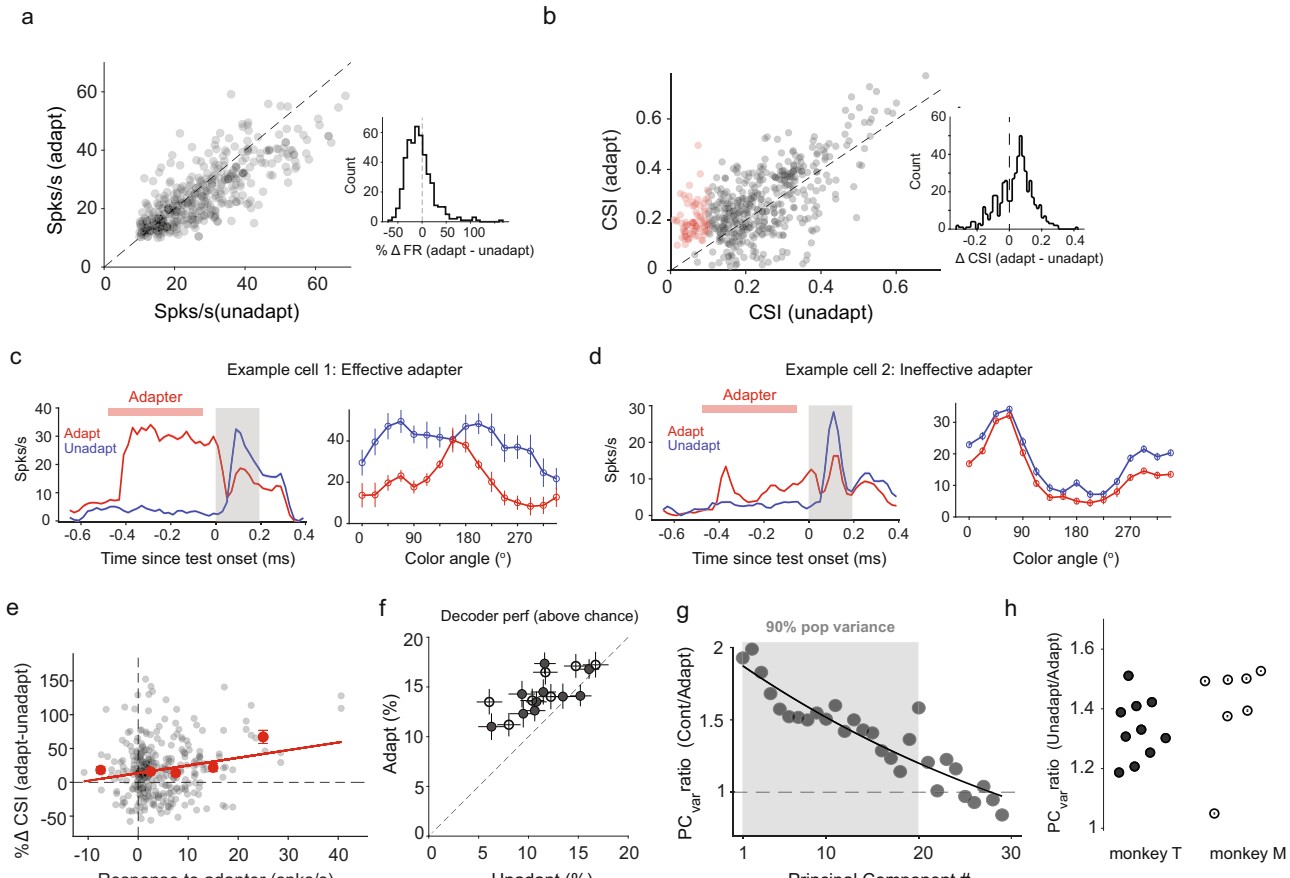

**Fig. 3 | Cross feature adaptation modulates stimulus coding. a** Neural responses ($n = 388$) to color stimuli for adapted and unadapted trials. Inset: Distribution of percent differences in responses (adapt − unadapt). **b** Color selectivity index (CSI) of neurons in adapted and unadapted conditions. Red circles represent color untuned neurons in the unadapted condition that gained significant color tuning after orientation adaptation. Inset: distribution ΔCSI values (adapt − unadapt, $ΔCSI_{mean} = 0.04$). **c** Post stimulus time histogram (PSTH) averaged over adapted (red) and unadapted trials (blue) of an example neuron (left) strongly activated by an effective oriented adapter and its associated tuning curve under both conditions (right). Red bar and gray shaded region represent presentation of adapter and test stimuli. **d** A different example cell showing weak activation by an ineffective adapter (left) and associated tuning curves under both conditions (right). Solid lines and shaded regions represent mean values and s.e.m respectively.
**e** Percentage change in CSI for neurons ($n = 283$) that were significantly tuned under both adapted and unadapted conditions as a function of the baseline

corrected response to the adapter. Black and red circles represent individual neurons and mean of binned values. Vertical bars and solid red line represent s.e.m and a linear fit to the binned values. **f** Mean decoder accuracy (%) above chance for classifying neighboring color stimuli (gray filled circle, $N = 10$ sessions monkey T; black open circles, $N = 7$ sessions monkey M) in both conditions. Vertical and horizontal bars represent s.e.m. **g** Ratio of the population variance captured in the trial-by-trial responses by PCs in unadapt and adapt conditions for an example session. Shaded region denotes the number of PCs that capture 90% of the population variance. Filled gray circles and solid black represent the ratio along each PC axis and an exponential fit. **h** Mean value of the ratio of variance explained by principal components for each individual session in monkey T (black filled circles, $N = 10$ sessions) and monkey M (black open circles, $N = 7$ sessions). Error bars (s.e.m) were evaluated by performing PCA on sub-sampled trials in each condition.

surprising since color and orientation were shown to be processed independently in anatomically non-overlapping modules in V4. Across sessions, we observed a small but significant decrease in peak firing rates (FR) after adaptation (Fig. 3a, Wilcoxon signed tank test, P < 0.05, $ΔFR_{adapt-unadapt} = −3.3 ± 1.4\%$, Fig. 3a inset). A significant proportion of color untuned neurons (18%) became significantly tuned (P < 0.05;

Rayleigh's test; Holm-Bonferroni correction for multiple comparisons; Fig. 3b) after exposure to a fixed oriented grating (see Supplementary Fig. 6 for results in each animal). In addition, adaptation increased the strength of color tuning (Fig. 3b) in roughly 45% of V4 neurons (statistical significance and strength of color tuning were examined using Rayleigh's test and Color Selectivity Index, CSI, see Methods). Across

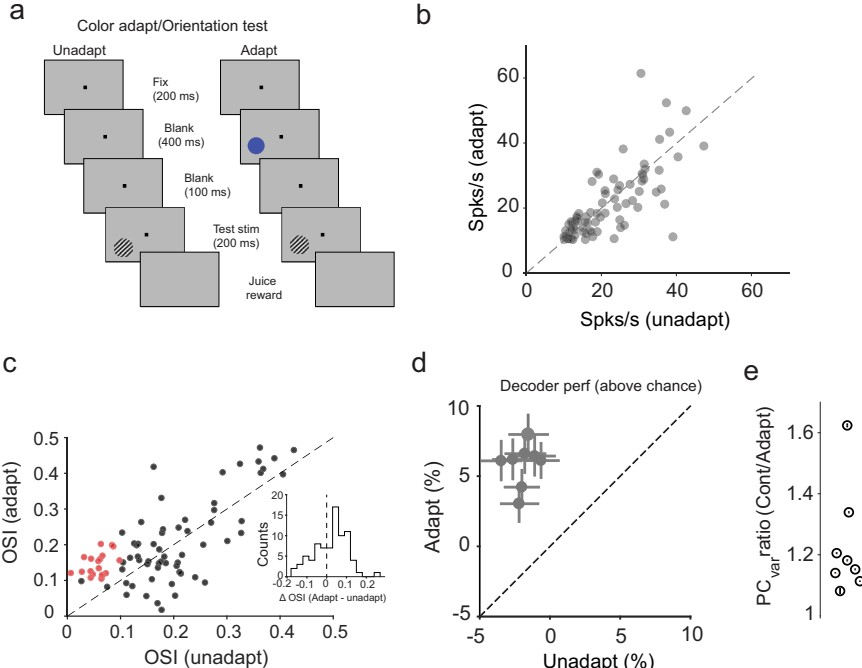

**Fig. 4 | Cross feature color adaptation effects. a** Schematic description of experimental design in case of cross feature color adaptation (color adapt/orientation test). **b** Mean peak responses of neurons ($n = 80$) to oriented stimuli for adapted and unadapted trials. Each gray circle represents a single neuron. **c** Orientation selectivity index (OSI) for neurons in adapted and unadapted conditions. Red circles represent neurons that were untuned to orientation in the unadapted condition but gained significant tuning after color adaptation; black circles represent all other neurons. Inset shows the distribution of change in CSI values (adapt − unadapt, $\Delta CSI_{mean} = 0.02$) calculated for all neurons. **d** Mean

decoder accuracy (%) above chance for classifying neighboring color stimuli based on the response of the neuronal population in each session (gray filled circle, 8 sessions) in unadapted and adapted conditions. Error bars represent s.e.m for the decoder accuracy over multiple iterations of the classifier. **e** Mean value of the ratio of population variance explained by principal components that account for 90% of the variance for each individual session in monkey T. Error bars (s.e.m) were evaluated by performing PCA analysis multiple times ($n = 100$) on sub-sampled trials in each condition.

the population of cells, we observed a net increase in the strength of color tuning strength after adaptation (Fig. 3b inset; $\Delta CSI_{adapt-unadapt} = 60 \pm 7\%$, $P < 0.001$, Wilcoxon signed rank test).

In general, we noticed that adapters that strongly activated neurons had a tendency to induce or enhance color tuning in neurons that were either untuned or weakly color tuned before adaptation (Fig. 3c, effective adapter). In contrast, adapters that only weakly activated neurons had a minimal impact on color tuning (Fig. 3d, ineffective adapter). Across the population, we found a significant correlation between how strongly an adapter activates the cell and the post-adaptation change in CSI (Fig. 3e; Spearman's correlation coefficient $r = 0.19$, $P = 0.0004$). Changes in tuning strength were also associated with changes in the temporal dynamics of color tuning. We calculated CSI as a function of time using a 200-ms window sliding by 5 ms, and defined tuning latency (TL) as the first time point when the neuron's tuning achieved statistical significance (Rayleigh's test). Tuning latency was significantly higher in the adapted compared to unadapted condition (Supplementary Fig. 7; Wilcoxon's signed-rank test, $P < 0.0001$, $\Delta TL_{adapt-unadapt} = 21.6 \pm 0.3$ ms). We found only modest changes in preferred color after adaptation ($\Delta\theta_{mean} = 1.4°$, Supplementary Fig. 8).

### Cross-feature adaptation improves population coding

Does cross-feature adaptation improve the ability of neuronal populations to discriminate between neighboring stimuli as observed in our free-viewing experiments? A classifier (linear discriminant analysis; see Methods) was trained to decode information from the responses of neural populations for neighboring color stimuli (separated by 22.5° on the color circle, Fig. 2a). Confirming our results in Fig. 1i, there was a

significant increase in mean decoder accuracy after adaptation (Fig. 3f; mean change$_{adapt-unadapt}$ = 5 ± 1%; $n = 17$ sessions, $N = 500$ iterations). This improved performance in decoding is surprising given the small, but significant decrease in overall firing rates post adaptation (Fig. 3a). This indicates that rapid adaptation to one visual feature improves population coding accuracy to distinguish between stimuli lying on an orthogonal feature axis.

To verify whether a similar mechanism to that occurring during free-viewing could explain the post-adaptation improved decoder performance during passive fixation, we repeated the principal component analysis on neurons' responses during the unadapted and adapted conditions (Fig. 3g, example session). We observed a significant reduction in variance of neural activity along principal component axes in adapted compared to unadapted conditions (Fig. 3h), i.e., population level activity was more decorrelated after adaptation, similar to the effects observed during free-viewing. This was surprising given that both the mean and the variance of the distribution of pairwise correlations were not significantly altered after adaptation (Supplementary Fig. 9). These results are robust with respect to the number of PC axes used, as we observed similar trends by considering only the first principal component axis (Supplementary Fig. 10a). This suggests that the decorrelation of neural responses across the population could be a general mechanism by which cross feature adaptation improves stimulus discriminability in both free-viewing and passively fixating animals.

### Cross-feature color adaptation

We further examined whether our results still hold when the adapting and test stimuli used in the experiments in Figs. 2–3, are swapped. Specifically, we analyzed neurons' responses to orientation after they

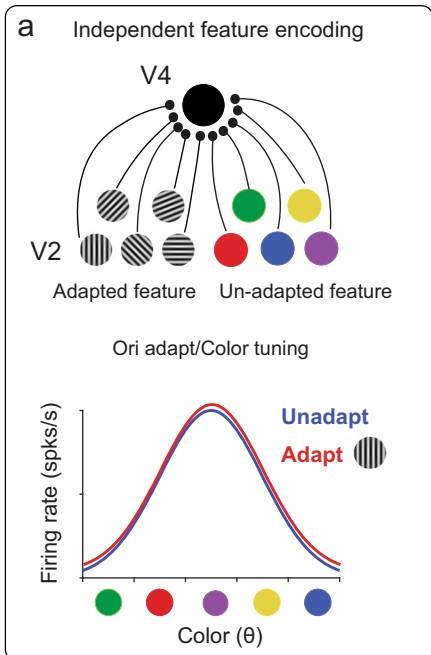

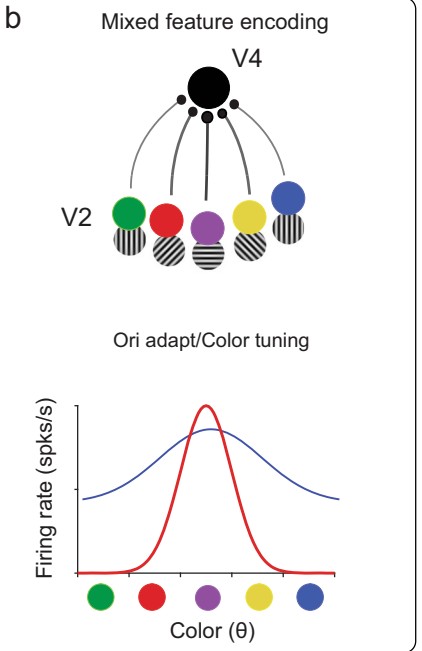

**Fig. 5 | Cross-feature adaptation involves mixed feature selectivity of inputs to area V4. a** Top: A target V4 neuron (solid black circle) receives heterogeneous orientation and color inputs from upstream neurons (e.g., in area V2) via anato-mically non-overlapping/distinct set of channels for each feature (independent feature encoding). This neuron is adapted to a vertical grating (orientation adap-tation) and its tuning to color (unadapted feature) is tested. Bottom: Schematic representation of tuning curves before (blue) and after adaptation (red) consistent with independent feature encoding, i.e., adaptation to oriented gratings does not affect color tuning. **b** Top: In contrast to **a**, a different V4 neuron receives both orientation and color inputs through the same synapses (mixed feature encoding). Some synapses are stronger than others (thick black line compared to faint gray lines) giving rise to broad band tuning for either feature. Bottom: Schematic depiction of tuning for this neuron before and after adaptation showing sharpen-ing/emergence of tuning due to adaptation-induced modification of specific synapses (via a reduction of flank responses).

were adapted to color while leaving all other task parameters unchanged (Fig. 4a; $N = 7$ sessions in monkey T). When using this new stimulus set, we found a small but significant decrease in peak responses to color stimuli after orientation adaptation (Fig. 4b; $\Delta FR_{adapt-unadapt} = -3.3 \pm 2.9\%$; $P = 0.02$, Wilcoxon signed-rank test). Similar to our original experiments, cross-feature color adaptation rendered a significant proportion of cells (18%) orientation-tuned despite an initial, pre-adaptation, lack of tuning (Fig. 4c). Furthermore, 28% of the orientation-tuned neurons exhibited a significant shar-pening post adaptation ($n = 123$ neurons; $P < 0.05$, Wilcoxon sign rank test, Fig. 4c inset). Cross-feature adaptation yielded an increase in decoder performance to distinguish between pairs of neighboring orientations (separated by 11.25°, Fig. 4d; mean increase: $7.7 \pm 0.5\%$; $n = 8$ sessions). Similar to the previous effects, we observed a sig-nificant reduction in the population variance along principal compo-nent axis in adapted compared to unadapted conditions (Fig. 4e, Supplementary Fig. 10b). Taken together, these results strengthen our initial findings that cross-feature adaptation during free viewing improves the ability of neuronal populations to distinguish between nearby stimuli lying on an orthogonal feature axis.

## Discussion

Natural scenes are characterized by a wide range of features, among which color and orientation represent two major components. While exploring natural scenes, during successive visual fixations, neurons are often exposed to image patches dominated by oriented signals, while subsequent fixations could land on distant image patches where color is the primary feature. It is widely believed that rapid adaptation during natural viewing could play a role in the efficient coding of diverse features (orientation, color, etc.) present in natural scenes[6]. However, this views has emerged from passive fixation experiments in which stimuli varied along a single feature axis. Therefore, whether

adaptation to a single feature influences neurons' response to a widely different feature (cross-feature adaptation) so as to influence the accuracy of stimulus encoding has remained unknown.

To our knowledge, our study is the first investigation system-atically examining the effects of cross-feature adaptation using a novel free-viewing paradigm whereby animals voluntarily explored stimuli consisting of multiple features (color and orientation) to show that the accuracy of visual cortical populations to discriminate stimuli is improved after cross feature adaptation. In addition, we used more controlled fixation conditions to characterize in greater detail the effect of cross-feature adaptation on neuronal tuning curves and population discrimination performance. Our most surprising finding is that a substantial fraction of visually-responsive neurons that were either untuned or poorly tuned for color or orientation became tuned after adapting to stimuli lying on an orthogonal feature axis, which may lead to a major revision of sensory adaptation and its neural underpinnings. Indeed, untuned neurons are usually ignored in most studies of cortical function, yet we demonstrate that they could play a significant role in the adaptive coding of sensory inputs. Furthermore, the remaining neurons, tuned to color or orientation, significantly improved their tuning after cross-feature adaptation.

What type of mechanism could possibly explain these adaptive cross-feature effects unreported in previous studies? Our results indicate complex feedforward and intracortical connectivity between cross-feature tuned V4 neurons. While orientation and color modules are believed to lie in separate functional domains, some V4 neurons could receive feedforward and intracortical inputs from a population of cells jointly encoding orientation and color signals[21–23]. Although these inputs may be insufficient to warrant a sharp tuning to cross-feature stimuli, neurons can nonetheless acquire orientation or color selectivity after adaptation provided that the cross-feature inputs are differentially weakened after the brief exposure to the adapting

stimulus. This indicates a higher degree of specialization of inputs to neurons in mid-level visual cortical areas than currently assumed.

More specifically, the prevalent feedforward model of feature encoding in V4 predicates that signals corresponding to orthogonal features, such as color and orientation, are transmitted via anatomically separate channels (independent feature encoding model, Fig. 5a, top). However, this model would predict that adaptation to one feature (e.g., orientation) will not alter the strength of inputs carrying information about the other feature (e.g., color). As a result, while orientation tuning of the target neuron is likely to change, the color tuning properties will remain unaffected (Fig. 5a, bottom). Alternately, the inputs to V4 neurons could simultaneously carry a broad range of color and orientation signals (mixed feature encoding model). For instance, Fig. 5b, top, illustrates a V4 neuron receiving mixed weakly tuned color and orientation signals. However, adaptation to a vertically oriented grating will specifically weaken synapses associated with the adapting orientation, which in turn will modify responses to the set of color signals (orthogonal feature) transmitted via the same set of synapses (Fig. 5b, bottom). The emergence of tuning to a specific color/orientation in untuned neurons depends on the heterogeneity of input signals for the adapted feature. For example, if feedforward orientation signals are similar across multiple inputs, adapting to that orientation will lead to a uniform reduction of neuronal responses across all colors instead of a selective decrease. Such an input structure would not lead to the post-adaptation emergence of tuning reported in our study. However, in realistic neural networks, the structure of feedforward inputs to cortical neurons is likely to be diverse, which is probably the main reason why the adaptation effects reported here are heterogeneous, ranging from emergence of tuning to loss of, or unchanged, tuning. Notably, despite this diversity, there was an overall significant increase in orientation and color tuning strength after adaptation and an improvement in stimulus discriminability through the decorrelation of population responses. Thus, specificity of rapid plasticity in neurons coupled with mixed feature selectivity could provide a mechanism for the observed changes in tuning. Our findings indicate a higher degree of specialization in the structure of inputs to mid-level visual cortical neurons beyond that proposed by the independent encoding models reported in previous studies. In fact, neurons with mixed feature selectivity (joint encoding of both color and orientation signals) have been found in macaque V1[23] which could be the source of mixed feature inputs to downstream areas.

Free-viewing a multi-stimulus patch presents a distinct set of challenges as saccades are voluntary (without a fixed trial structure) and the locations of neurons' receptive fields change along with the gaze of the animal. We addressed these challenges by calibrating eye movements and thresholding eye velocity to extract pseudo trials in the unadapted and adapted conditions, and eliminated trials where the receptive fields of neurons covered more than a single stimulus or were close to the stimulus boundaries or the edge of the monitor. While free-viewing allowed us to examine rapid adaptation across features under more naturalistic conditions, we complemented the free-viewing approach with passive fixation experiments for a more detailed characterization of changes in tuning properties. Together, this complementary approach revealed that cross-feature adaptation is a general feature of visual cortical circuits that manifests both in naturalistic and fixed gaze viewing conditions.

A key result in our study is that rapid cross-feature adaptation improves population coding accuracy. We propose that the primary mechanism responsible for this adaptive increase in population coding relies on a reduction in shared variability among neurons. Indeed, theoretical work has shown that changes in correlated activity in neurons influence stimulus encoding in sensory areas[24–27]. Furthermore, using principal component analysis, recent work[21,22] has shown that learning and decision making are associated with changes in shared variability at the population level. Employing a similar approach, we found that cross-feature adaptation leads to a significant decrease in shared variability along principal component axes that explain most of the response variance (around 90%). This finding was consistent across free-viewing and passive fixation experiments using color and orientation adapters. Our population analysis indicates that rapid cross feature adaptation causes an overall decorrelation in the structure of trial-by-trial population activity. From a functional standpoint, this implies that exposure to an oriented stimulus prepares the color encoding system by decreasing variance across the neural population, thereby enabling a rapid emergence of color tuned responses during successive fixations (hence increasing coding accuracy). Interestingly, rapid adaptation decorrelates neuronal responses to different features (color and orientation) in qualitatively similar ways. Additionally, analysis of iso-feature adaptation data (adapter and test stimuli belonging to the same feature), revealed similar qualitative changes in tuning strength, discriminability and population response variance (Supplementary Fig. 11). This indicates that decorrelation could be a general mechanism at the population level to efficiently encode a wide variety of sensory stimuli after adaptation[28].

Surprisingly, there is a complete lack of studies on either rapid or prolonged adaptation in the cross-feature domain with respect to changes in stimulus discriminability. Previous studies have revealed cross-orientation[29] and cross-sensory interactions[30,31], however the role of such interactions on stimulus discriminability has not been explored. Psychophysical evidence of cross-feature adaptation, although of a very different kind than that presented in our study, has been observed in the form of the McCullough effect[32]. Although this color aftereffect due to long-term adaptation (2–4 minutes) implies interaction between features, it is not associated with an improvement (at any time scale) in the discriminability of features, as reported in our study. Furthermore, a more recent study[33] has examined the effects of adaptation (~5 s) to contrast and luminosity on the encoding of oriented stimuli to report an improvement in discrimination thresholds post adaptation. However, unlike color/orientation encoding which are believed to be represented by anatomically distinct modules, features such as contrast and luminance are typically represented by the same population of neurons as that encoding for orientation stimuli. Our findings in single neurons and populations are likely to motivate future psychophysics experiments using both synthetic and natural stimuli to directly test the perceptual effects of cross-feature adaptation. Additionally, although we characterized the cross-feature adaptive interactions using well parametrized synthetic stimuli (orientation and color), our work sets the stage for future endeavors to explore rapid adaptation using more complex stimuli found in natural scenes. Adaptive changes in population coding accuracy similar to those shown here may exist in other brain areas. Therefore, examining the adaptive properties of the neural code across the visual system, or other systems, will likely provide important clues about the link between neural population activity and perception during naturalistic viewing.

## Methods

All experiments were performed in accordance with protocols approved by the U.S. National Institutes of Health Guidelines for the Care and Use for Experimental Procedures and the Institutional Animal Care and Use Committee (IACUC) at the University of Texas Health Science Center at Houston. Data presented in this study was collected from three adult male rhesus monkeys (*Macaca mulatta*; T: 14 years old, 13 kg; M: 10 years old, 10 kg and R: 11 years old, 12 kg). A titanium head post was surgically implanted in the medial frontal region with the help of multiple anchor screws. After a recovery period of 4 weeks, both animals were trained for a month on visual fixation tasks (to be used later for the recordings) involving at least 1000 trials per session. After the monkeys learned to complete multiple sessions in a single day, we implanted a 96-channel Utah array in monkey T and R and a

64-channel Utah array in monkey M in area V4 (left hemisphere in all animals). Coordinates for craniotomies were estimated based on locating the superior temporal sulcus (STS) and the lunate sulcus by comparing MRI images from the animals to brain atlases. During surgery, the grooves corresponding to the STS and the lunate sulcus were used to guide the implantation of the array. Arrays were roughly implanted at the crown of the pre-lunate gyrus. Post-surgery, animals went through a 2-3 week recovery period and additional training for re-acclimatization with previously learned fixation tasks before we started recording.

## Visual stimuli for single unit recordings

Visual stimuli were presented on a gamma corrected CRT monitor (HP p1230). To measure color tuning we used a set of 16 equiluminant colors ($9.35 \pm 0.01$ cd/m²) spanning the full color gamut of the monitor and presented at the maximum saturation allowed by the monitor[20]. These 16 colors were uniformly spaced (in steps of 22.5°) when plotted in the CIELUV color space[34,35] (Fig. 2a) which is designed to be perceptually uniform. L represents the luminosity, and u, v represent the chromaticity coordinates in a two dimensional perceptually uniform color space. Luv coordinates were measured for each color and the gray background using a Tektronix photometer (J17 Lumacolor) before the start of each recording session. These colors were presented on a neutral gray background designed to have the same luminosity as the color stimuli (see Supplementary Table 1). Grating stimuli consisted of 16 orientations spanning 0° to 180° in steps of 11.25°. The mean luminosity of the gratings was matched to that of the color patches and the gray background. Visual stimuli were presented binocularly; eye tracking was performed for only one of the eyes.

## Behavioral task

Two monkeys were trained to fixate on a small point (0.2 deg) within a small rectangular 1-deg window at the center of a cathode ray tube (CRT) monitor while remaining head fixed. If at any point during the trial, eye position exceeded 0.25 deg outside the boundaries of the rectangular box, then the trial was automatically aborted. Animals were rewarded with juice at the end of each trial in which fixation was successfully maintained for the entire duration of stimulus presentation. Eye movements were monitored throughout the recording session using an infrared eye tracking system (EyeLink II, SR Research) at a 1-kHz sampling rate. Stimulus presentation was recorded and synchronized with the neural data using a programmable Experiment Control Module device (FHC Inc.).

## Electrophysiological recordings

We recorded extracellular activity as action potentials and local field potentials simultaneously from all 96 channels (monkey T, R) and 64 channels (monkey M) of three chronically implanted Utah arrays (Blackrock Microsystems) while animals performed passive fixation tasks. The interelectrode spacing in these arrays was 400 μm. Data was recorded at a sampling rate of 30 kHz using a Cerebus Neural Signal Processor (Blackrock Microsystems, LLC). Spike waveforms above threshold (~4 sd above the amplitude of the noise signal) were saved and sorted post data acquisition using Plexon's Offline Sorter. Spike waveforms were manually sorted with Plexon's offline sorter program using waveform clustering parameters such as spike amplitude, spike width, timing of the valley and peak. Units that formed well separated clusters in principal component space were identified as single or multi units. Units that had more than 2% of their post sorted spikes within the refractory period (2ms) were classified as multi units and were eliminated from the analysis[36]. The remaining single units were subsequently analyzed using custom scripts in MATLAB. We treated each recording session performed on a particular day as an independent session, which is a common approach adopted by multiple labs[20,21,37–39]

as it is difficult to determine whether the same units were recorded on subsequent days over the course of several months.

## RF mapping

To map the RFs of the single units we divided the right visual field into a 3 × 3 grid consisting of 9 squares with each square covering 8 × 8 degrees of visual space. The entire grid covered 24 × 24 degrees of visual space. Each of the 9 squares was further subdivided into a 6 × 6 grid. In each trial, 1 out of the 9 squares was randomly chosen and RF mapping stimuli was presented at each of the 36 locations in a random order. The RF mapping stimuli consisted of a reverse correlation movie with red, blue, green and white patches (~1.33 degrees each). A complete RF session comprised of 10 presentations of the RF mapping stimuli in each of the 9 squares forming the 3 × 3 grid. We averaged the responses over multiple presentations to generate the RF heat maps. RF mapping was done at the beginning of each recording session as it was impossible to track the same neurons over the course of the recordings which lasted several months. The position and size of the target stimulus for examining tuning was chosen each day so that it roughly covered the overlapping RFs of only a subset of neurons. Neurons whose RFs could not be covered completely by the chosen target were not included in the analysis. This ensured that the size of the target could be kept small so as to minimize surround stimulation.

## Free-viewing adaptation task

A four-panel static stimuli (Fig. 1e) was designed to evaluate cross-feature adaptation in a free-viewing paradigm. The upper right quadrant was a grey isoluminant control, the upper left quadrant was the adapter stimulus, and the lower quadrants were test stimuli. To assess responses to color after adaptation to an oriented grating, we used a grating adapter stimulus oriented at 135° with a spatial frequency of ~2.8 cycles/deg. Grating parameters were chosen to elicit responses from as many neurons recorded by the array. The hue angle corresponding to the 2 test stimuli were 45 and 67.5° (neighboring stimuli) and were placed in the lower left and right quadrants respectively. To assess response to oriented gratings following adaptation to color, we used a color adapter stimulus of 0 degrees of hue angle. The oriented gratings were 112.5 and 135° for the bottom left and right quadrants respectively. Before presenting the static task stimulus, eye position was first calibrated with respect to the monitor. The task stimulus was presented on the monitor for a total duration of roughly 1 hr in each session. A drop of juice was automatically given to the animal at regular intervals provided its point of gaze was on the stimulus monitor. Saccades and fixations were analyzed offline, and separated by a velocity threshold of 100 deg/s[40]. We extracted 'pseudo-trials' from the free-viewing recordings based on the sequence and location of fixations with respect to the task stimulus. Unadapted trials were defined as those in which the animal first fixated on the isoluminant grey quadrant (upper right), and then fixated on one of the two lower test stimulus quadrants. Adapted trials were defined as those when the animal first fixated on the adapter quadrant (upper left) and then fixated on one of the two lower stimulus quadrants. Fixation-related variables such as duration of fixations on test or adapter stimuli, velocity and direction of microsaccades, pupil size during fixations were analyzed to show that these variables are not significantly different in the adapted and unadapted trials. Additionally, only fixations for which the full RFs of neurons were confined to one quadrant and did not include the edge of the monitor boundaries or the boundaries separating the different stimuli were included in the analysis (Supplementary Fig. 4).

## Sobel filter analysis

We computed the orientation content of images by applying a Sobel filter[41–43] to each image. For each pixel we determined both the

orientation and the orientation magnitude by calculating the partial derivative of brightness in a 3 × 3 kernel. The orientation was obtained by calculating the arc tangent of the vertical component divided by the horizontal one. The orientation magnitude of the local gradient was calculated from the square root of the sums of the squares of the partial derivatives of the brightness in the vertical and horizontal directions. We then determined the orientation magnitude histogram of an image patch as the number of pixels at each particular orientation (collapsed to a 0°–180° in 5° bins) weighted by the magnitude of the gradient at that pixel. We subsequently calculated the mean orientation and orientation selectivity index (OSI; Fig. 1b) of image patches by extracting the Fourier components from the orientation magnitude histogram for each patch. The Sobel filter analysis was performed on sub-patches of the natural images in a 50 × 50 pixel window that was moved horizontally and vertically by 10 pixels to cover the entire image.

### Statistical model of visual exploration of natural scenes

We generated a distribution of saccade amplitude and direction (Supplementary Fig. 2) from the free-viewing experiments described above. Based on these statistics, we generated pseudo-saccades from randomly chosen starting points on a natural image battery ($n = 940$) from the McGill calibrated color image data base (McGill Vision research; http://tabby.vision.mcgill.ca/) to simulate unrestrained visual exploration. We analyzed a total of 282,000 saccades across all images (300 saccades per image). The size of image patches was chosen based on mean receptive field size of our V4 neurons. For every pseudo-saccade connecting two patches of a natural image, we quantified the strength of orientation and color signals. Orientation signals were calculated by computing an orientation selectivity index (OSI) using a Sobel filter (see Methods). Color signals were quantified by analyzing the distributions of the red, green and blue channel values across all pixels within a patch (values lying between 0 to 255). A patch was considered to be 'colored' if it met the following requirements: (a) each of the peaks of the R, G and B channel distributions were not located within a threshold $\Delta$ (set at 30) to either 0 or 255 (black or white patches). (b) The difference in the peak locations for each channel were not closer than $\pm\Delta$ (to rule out gray patches). (c) The patch did not have significant orientation signals based on the Sobel filter analysis. Patches were considered 'oriented' if there was significant tuning (Rayleigh's test, pval <0.05) based on Sobel filter analysis and did not have color information based on the criteria presented above. To ensure that patches with no significant orientation tuning did not simply contain multiple orientations (multiple peaks in their OSI distribution), we eliminated from the analysis those patches containing more than one peak in their OSI distribution, where peaks were defined as local maxima in the OSI distribution (>3 s.d. above the mean). For each image we analyzed 300 saccades to calculate how often saccades connected two patches with dissimilar/similar features.

### Color and orientation tuning

To determine the color/orientation tuning of V4 neurons we calculated the tuning curve from multiple presentations (N ~ 30/per stimuli) of the chosen stimulus feature (color patches/gratings). For each such tuning curve, we calculated the color/orientation selectivity index (CSI/OSI) in the following way: $CSI = \frac{|\sum_i r_i e^{\theta_i}|}{\sum_i r_i}$ where $r_i$ is the response to a particular color/orientation with angular position $\theta_i$ on the color/orientation space. Responses $r_i$ were vectorially summed and then normalized by the sum of the responses for all colors. To control for the effect of noise on estimation of tuning curves, neurons with peak firing rates <10 spks/s were removed from the analysis. CSI lies between 0 and 1, where a value of 0 represents no tuning and a value of 1 indicates that the neuron is highly tuned for a specific color. To evaluate whether the neuron's tuning was statistically significant, we used the Rayleigh test for non-uniformity in circular data[9,20,44]. Rayleigh's test is a statistical test used to determine whether a circular distribution (which in our case is the circular tuning curve with firing rates) has a preferred direction, i.e., a preferred color the neuron is tuned to ($P < 0.05$). The null hypothesis is that the circular tuning curve represents a uniform circular distribution and no preferred direction exists ($P > 0.05$). We evaluated CSI and the p-values of the tuning curves in sliding windows (size 200 ms, sliding by 5 ms) from the stimulus responses measured separately for unadapted and adapted trials. We performed Holm-Bonferroni correction for multiple comparisons on the p-values. To evaluate the preferred color/orientation (PC/PO) we selected the tuning curve with the highest CSI/OSI which passed the Rayleigh test for non-uniformity ($P < 0.05$). This tuning curve was then used to calculate PC/PO in the following way: $PC = \tan^{-1}\left(\frac{Im(\sum r_i e^{\theta_i})}{Re(\sum r_i e^{\theta_i})}\right)$ where $Im$ and $Re$ stands for the imaginary and real parts of the complex sum. We evaluated tuning latency (Fig. 3f) as the first time point where the neuron exhibited significant tuning to a feature ($p < 0.05$, Rayleigh test). To eliminate the effect of noise in evaluating tuning strength and preferred stimulus we eliminated neurons with peak firing rates <10 Hz. Circular statistics was computed using the Circ-Stat Toolbox[45] for MATLAB.

### Adaptation task (passive fixation)

In adaptation trials an adapter of a particular feature (color or orientation) preceded the presentation of a tuning test stimulus of another feature. The adapter was presented for 400 ms and after a 50 ms delay (gray screen), a randomly chosen tuning stimulus was presented briefly for 200 ms. In unadapted trials the presentation of the tuning stimuli was not proceeded by an adapter. A complete session involved randomly interleaved adapted and unadapted trials (no adapter). The adapting stimulus was chosen so that it activated a majority of the recorded neurons in a session. We recorded close to 1000 trials in these sessions split roughly equally between unadapted and adapted conditions. The duration between the last frame of the tuning stimulus and the first frame of the adapting stimulus in the next trial was at least 4.6 s which allowed sufficient time for the responses to return to baseline.

### Noise correlation analysis

Spike count correlations between pairs of neurons was evaluated using the Pearson's correlation coefficient defined as follows:

$$r_{sc} = \frac{\sum_{k=1}^{N}(r_{ik} - r_j)(r_{jk} - r_j)}{N\sigma_i\sigma_j} = \frac{\sum_{k=1}^{N} r_{ik}r_{jk} - r_i r_j}{N\sigma_i\sigma_j}$$

Where N is the number of trials, $r_{ik}$ is the firing rate of neuron $i$ in trial $k$, $r_i$ is the mean firing rate (averaged over trials) evaluated in a 200 ms window aligned with test stimulus onset and matching the duration of test stimuli presentation, and $\sigma_i$ is the standard deviation of the responses of the neuron $i$. Neurons with peak firing rates < 5 spks/s were not included in the noise correlation analysis. Additionally, trials in which the absolute values of the z-scored firing rates of either neuron exceeded three times the mean were eliminated from the analysis. Finally, we averaged the noise correlation across all stimuli for a particular feature to report mean correlated activity in neuronal pairs. Noise correlations were calculated for all pairs both before and after adaptation. In case of the free-viewing recordings, spike counts were evaluated for the duration of each fixation on the test stimuli.

## Principal component analysis of population activity

We performed principal component analysis (PCA) on a trial-by-trial basis for the entire population of neurons[21,22], in both unadapted and adapted conditions for each experimental paradigm. Principal Component analysis was performed on responses in each trial (e.g., firing rates during fixations) of all recorded neurons in the unadapted condition, i.e., PCA was performed on a M x N matrix where M denotes trials and N the number of neurons. To control for the effect of high variance in single neuron responses (outliers) dominating the PCA, we removed neurons whose variance exceeded 2 std above the mean variance of all the neurons in a session. We then projected the trial-by-trial activity of the same neurons during the adaptation condition onto the PC axis obtained for the control condition and examined the ratio of the variance explained by each principal component in the control and adaptation condition by focusing on the mean ratio for those PCs that captured 90% of the variance of the population response in the control condition. To obtain error estimates for ratio values, PCA analysis was repeated 100 times with a different set of sub-sampled trials (same number) in unadapted and adapted conditions.

## Decoder analysis

We used linear discriminant analysis[46–48] to examine whether cross-feature adaptation enhanced the ability of neural populations to discriminate between nearby stimuli. We analyzed decoder accuracy for pairs of nearest neighbor stimuli in feature space, i.e., separated by 11.25° in the case of gratings and 22.5° in the case of color stimuli. The decoder was trained on spike counts evaluated in 200 ms (0–200 ms post tuning stimulus onset) for a subset of the trials. In the case of free-viewing recordings, spike counts calculated for the fixation window were used as inputs to the decoder. The classifier was trained on 70% of the trials and performance was tested on the remaining 30%. We repeated this classification process 500 times with different training and testing sets drawn randomly from the trials from a session. Decoder performance was averaged over the number of iterations and evaluated for both unadapted and adapted conditions. We subtracted chance values of decoder performance for a pair of stimuli (50%) from the actual values and thus reported decoder accuracy values above chance for both free-viewing and passive fixation experiments.

## Statistical analysis

Quantification and statistical test for tuning was performed using the Rayleigh test implemented with the CircStat Toolbox[45]. Correlations were quantified using either Pearson's correlation or Spearman's rank correlation to account for linear/non-linear trends as applicable. We used the non-parametric Wilcoxon's signed rank test (two tailed) to quantify whether distributions had medians significantly greater than or less than zero. In case of two distributions with unequal sample size we used the Wilcoxon rank sum test to examine the statistical significance of the difference in their medians.

## Reporting summary

Further information on research design is available in the Nature Portfolio Reporting Summary linked to this article.

## Data availability

Source data used to generate the main and Supplementary figures are provided as a Source Data file and can be accessed here https://zenodo.org/record/7411296. The raw data that was analyzed to generate the findings in this study are available from the corresponding author upon request. Source data are provided with this paper.

## Code availability

The custom written software used for analysis are available from the corresponding author upon reasonable request. The CircStat toolbox used to perform circular statistical analysis is open source software and is available for download from https://www.mathworks.com/matlabcentral/fileexchange/10676-circular-statistics-toolbox-directional-statistics.

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

## Acknowledgements

This work is supported by NEI grant 5 R01 EY031588.

## Author contributions

S.N., R.M. and V.D. designed the experiments. S.N. and R.M. performed the experiments. S.P. wrote the scripts for stimulus presentation using Psychtoolbox for the passive fixation experiments. R.M. wrote scripts for stimulus presentation for the free-viewing experiments. S.N. analyzed the data with guidance from V.D. S.N. and V.D. wrote the manuscript.

## Competing interests

The authors declare no competing interests.
