## [Peer Review File · Nature Communications]

Adaptive coding across visual features during free-viewing and fixation conditionsREVIEWER COMMENTS

Reviewer #1 (Remarks to the Author):

The manuscript "Adaptive coding of naturalistic signals in visual cortex" by Nigam, Milton, Pojoga and Dragoi investigates the question of cross-stimulus adaptation of V4 neurons. Despite the title and Figure 1a, this is not a study with naturalistic stimuli, which is disappointing.

The key results are that a decoder increased performance to dissociate between color stimuli when neurons were adapted to orientation and vice-versa. The authors then link this effect to neuronal tuning refinement. They also indicate changes in noise correlations distributions across neuronal population (but average noise correlation level remains the same) in the freely viewing condition, but this is absent in the more controlled passively viewing condition. While the authors analyse a large data set and some of the observations on cross-stimulus adaptation are interesting, especially the decoder, the a large number of included neurons responded poorly to the presented stimuli and there are also some methodological issues. Given the variable results between the two experiments and within each data set, it remains unclear whether the findings on cross-stimulus adaptation can add significant new insights into the neural mechanisms of adaption of visual cortical neurons beyond what we know from some of the careful studies already done in primate V1.

Major concerns:

1) The title, Figure 1a, and introduction suggest an investigation into the important question of adaption when subjects view actively a complex visual scene. However, the authors then proceed to investigate increasingly reduced paradigms. The stimuli are all traditional, simple colour or grating stimuli. While in the first paradigm (Fig.1), the animals are still freely viewing the stimuli, the main bulk of the paper covers a very traditional experiment with passively fixating animals while stimuli are projected on the receptive field (Fig.2).

2) Placement of stimuli in the receptive fields and firing rates during freely viewing (Figure 1).

2.1) Figure 1e shows the stimulus configuration and example landing positions of saccades in the adapted and unadapted condition. Figure 1d shows the receptive field (RF) positions and sizes for the array recordings (RF: X= +4°-5°, Y=-4°, size = ca. 4°x3°).

Taken together these figures indicate that at least some receptive fields in the "adapt" condition did span the two colour stimuli (red saccade landing points 2-3° from the dividing line). Furthermore, some of the lower saccade starting points in the "unadapted" condition would have receptive fields reaching

easily into the colour zone underneath during, so the responses would be for one colour stimulus against another.

It would be helpful to plot the receptive field positions not just the saccade landing points and show that for all included data the receptive fields are fully inside the relevant stimulus boundaries.

2.2) The authors control for microsaccade sizes but not endpoint directions and mean direction of microsaccades that could maybe impact this central results.

2.3) Figure 1g shows that the bulk of the recorded neurons did not respond well to the tested stimuli (<10 spikes/s). Thus, it is unclear what contribution, if any, a large number of the analysed neurons would make in a related perceptual task. Also, low firing rates can lead to spuriously high or highly variable interneuronal correlations. There should be a reasonable inclusion criterion for analysing the data further (e.g. ≥ 10 spikes/s).

Also, since in the first experiment, reward was given at regular intervals unrelated to task stage, this could have affected neuronal responses as well.

3) The authors indicate changes in noise correlations distributions across neuronal population (Fig. 1h) in the freely viewing condition, which are interesting, but were unfortunately this was not confirmed in the more controlled passively viewing condition. This suggests they are not a general mechanism for cross-stimulus adaptation.

4) Another interesting feature in the data are the neurons that show tuning in the adapted state, but not in the unadapted state (Figure 2d). However, if one looks at Figures 3b and 3c, there seems also to be a sizeable number of cells that show the opposite behaviour. Given that most of the analysed cells show a low OSI or CSI, this suggests that we are dealing with a noisy data set, where by chance some cells show this behaviour. It might be helpful to restrict the analysis to cells with a clear response and tuning to the stimulus and show the effect in this more controlled data set.

The authors show an increase in decoding accuracies that is about the “net gain” in populational encoding. I think it would be informative to look at the dynamics of this gain of information and changes in tuning. For example to split session trials between beginning and end of the session and compare decoding performance and tuning curves of the neurons in the adapted conditions, if the authors are correct, I would expect that during phases of the experiment, even in the adapted state when the tuning is stronger to have more effect on the decoder. Thus, a link between the two observations could be demonstrated.

5) For the 2nd experiment, the authors suggest that pairs neurons change their interneuronal correlations (from positive to negative and vice versa) between the unadapted and adapted state (Figure 3h). Unfortunately, the combined distribution does not look bimodal, but rather Gaussian. Since adapted and unadapted trials have been collected in an interleaved fashion for each cell pair, it seems possible that this result appears randomly by dividing the trials of the cell pairs and then looking at the extremes of what is a single distribution. A random permutation test on could clarify this questions.

6) There are neurons in V1 that have a range of orientation and colour selectivity and their combination (Johnson, Hawken, Shapley J Neurosci 2008; Garg, Li, Rashid, Callaway, Science 2019; cited by the authors). And V1 neurons have been shown to have different dynamics on different timescales for specific and non-specific inputs from just outside the classical RF, which is also stimulated here (Henry, Jazayeri, Shapley, Hawken eLife 2020). Given that in the 2nd experiment adaption show the expected decline in firing rate with adaption (figures 3i and 4e), it seems likely that the V4 responses could be explained by inheritance of signal changes from V1 from neurons that show joint encoding of orientation and colour. A distinct clear model of the underlying neural mechanisms V4 is not explicitly offered. The manuscript would benefit from theoretical underpinning and the proposal of a mechanistic model for the cross-feature adaptation.

Minor points:

7) It would be helpful to the reviewer to have page and /or line numbers.

8) The main results should be confirmed separately for each monkey in the supplement.

Reviewer #2 (Remarks to the Author):

This study addresses an important question regarding neuronal adaptation during natural viewing conditions. That is, during natural eye movements, the statistics of each viewed patch may differ significantly, entailing stimulation by features of different feature spaces (e.g. orientation vs color content). However, few studies have systematically examined the effects of such cross-feature activations on neuronal response.

To address this question, the authors have designed an 'unadapt (within-feature)' and an 'adapt (cross-feature)' viewing paradigm, in which monkeys are required to look from e.g. a blank gray quadrant to a color quadrant (within-feature) or from an oriented quadrant to a color quadrant (cross-feature). They find adapt conditions lead to emergence or enhancement of tuning for cross-feature conditions. This suggests (1) that there are interactions between neurons in what is considered different parameter spaces and (2) that these cross-feature specific enhancements may be important for signalling locations of feature change in the environment during natural viewing. The study is well conducted, the paradigm novel, and the results interesting. But there is a concern about the assumptions of 'cross-feature', making the significance unclear.

Primary comments

Truly cross-feature? Much of the significance of the study hangs on what exactly is cross-feature. I feel this needs to be more deeply examined. That is, stimuli can be characterized as in a 'color/surface' category or an 'orientation/achromatic' category. However, each cell's preference in V4 may not be so simple. Whether each cell's response deviates from its 'feature category' then depends on some quantitative characterization of its multi-feature tuning space.

Does one really need to invoke the cross-feature characterization or will any salient change lead these observed changes?

Cell types? What is unclear to me is, for the cells that reveal enhanced cross-feature tuning, are these cells hue broadband and hue tuned cells that become enhanced in their preference, or are they oriented cells that do not typically have color tuned response? I think the meaning of this enhancement would be quite different depending on whether this is observed in e.g. a color domain or an orientation domain. That is, every cell can be viewed as a multi-dimensional entity with some bias towards a baseline state, one which is predominantly biased towards 'orientation' in orientation domains and toward 'color' in color domains. Evoking color tuning to a color broadband cell may not be cross-feature, while evoking this in an orientation cell would qualify as cross-feature. Note that this is an oversimplification, as there are many cells in V4 of great complexity. Did the authors try to classify, in their paradigm, whether cells are color selective/orientation non-selective, orientation selective/color non-selective, color and orientation selective,...?

Discussion weak: There is a lack of discussion regarding what mechanisms/circuits might underlie their observations. This cross-feature interaction is reminiscent of cross-orientation effects whereby an oriented stimulus enhances the cells of the preferred orientation and relatively suppresses those of orthogonally oriented cells. This push-pull effect effectively sharpens orientation tuning and 'tightens' the circuit. Do the authors think their findings might be explained by something analogous? What do the longer latencies mean? Why the repulsive shift?

Minor comments

I did not find Fig 3e, 3f compelling.

Reviewer #3 (Remarks to the Author):

This paper describes the effects of cross-feature adaptation in area V4 of the monkey. While the results may be interesting and important, there are several problems with this paper that make it very difficult to evaluate in its current state. My comments are listed below.

Main comments

1. Why is cross-feature adaptation important and why is it useful? In general, adaptation is useful because it adjusts the sensitivity of neurons in response to the intensity of external stimuli. Why does the nervous system want to adjust its sensitivity between features, especially after saccades? Is there any theoretical or psychophysical literature on this point?
2. The results report only on cross-feature adaptation. This greatly hampers the interpretation of the results. There is a need to compare inter-feature adaptation with cross-feature adaptation. For example, what are the differences in color tuning and decoder performance between color adaptation and orientation adaptation?
3. More explanation of the experimental protocol and analysis is needed. In particular, it is very important to know where the receptive fields of the neurons were located relative to the boundaries of the adapting, control, and test stimuli in the free-viewing condition. The method corresponding to Figure 1e, where control and adaptation trials are sorted based on the start and end points of the saccade, only indicates that the receptive field was within the boundaries of the monitor. However, the receptive field is located to the lower right of the fixation point, and it is unclear whether the receptive field is restricted to the adapting and control stimuli in adaptive and control trials, respectively. Furthermore, can the direction of the saccade affect the neuron's response?
4. The discussion section is poor and needs to be more refined.

4a. Why does the decoder perform better in adaptive trials versus non-adaptive trials? The mechanism needs to be discussed. Can it be explained by changes in noise correlation? In general, decoding performance depends on the average noise correlation and not on the distribution.

4b. The results on noise correlation seem to be different between the free-viewing experiment and the fixation experiment. How can this difference be explained?

4c. What neural circuitry is responsible for the increase in the negatively correlated pairs and the decrease in the positively correlated pairs? What is the significance of this change in noise correlation?

Minor comments

1. The title is too general. The paper deals with cross-feature adaptation, and the experiment does not deal with naturalistic signals themselves.

2. Which hemisphere did you record from in each monkey? Is it the left hemisphere in all monkeys?

3. Which monkeys participated in each experiment?

4. How did you extract the "pseudo-trials" for the free-viewing task? What are the criteria?

5. In the free-viewing task, how was the time window for calculating spike counts for decoder analysis set?

6. Figure 1d. Why is the origin not the fixation point?

7. Figure 1j. Why is the decoder performance for unadapted stimuli barely above chance level? It seems inconsistent with the fact that many V4 neurons are selective for color and orientation.

8. Figure 1j. Where are the open circles? How many sessions were conducted on each monkey?

9. Figure 3. The title should be adjusted as in figure 4.

10. Figure 3b. What are the dotted lines at 0.1? If they do not represent statistical criterion, they should be omitted. Make the inserted figure larger.

11. Figure 3j. Is the data from one monkey? If the data are from different monkeys, use different symbols.

12. Figure 4d: Why is the format different from Figure 3h?

13. Figure 4f. Which monkey is this data from? Why is the decoder performance for unadapted stimuli lower than zero?

14. I don't think Supplementary Figure 1 is necessary. This has already been pointed out in Fig. 1a.

15. The legend in Supplementary Figure 2b is actually the legend for Supplementary Figure 2c, and the legend for Supplementary Figure 2b is missing.

Reviewer #1

Major concerns:

1) The title, Figure 1a, and introduction suggest an investigation into the important question of adaption when subjects view actively a complex visual scene. However, the authors then proceed to investigate increasingly reduced paradigms. The stimuli are all traditional, simple colour or grating stimuli. While in the first paradigm (Fig.1), the animals are still freely viewing the stimuli, the main bulk of the paper covers a very traditional experiment with passively fixating animals while stimuli are projected on the receptive field (Fig.2).

We used the term “naturalistic signals” in the context of two important limitations of previous adaptation studies. First, previous studies have exclusively focused on experimental paradigms involving restricted viewing in which stimuli were presented during passive fixation, thereby lacking the naturalistic conditions encountered during free-viewing. Second, the conclusions of previous studies originated from investigations of how the neural code adapts to a relatively narrow set of stimuli along a single feature axis, such as orientation, motion, or color. However, during a typical visual fixation, neurons can be exposed to image patches dominated by oriented signals, while subsequent fixations could land on distant image patches where color is the primary feature. Whether and how *cross-feature* adaptation influences the accuracy of stimulus coding remains unknown. We have now chosen a title that more accurately describes our study, “*Adaptive coding across visual features during free-viewing and fixation conditions*”.

2) Placement of stimuli in the receptive fields and firing rates during freely viewing (Figure 1).

2.1) Figure 1e shows the stimulus configuration and example landing positions of saccades in the adapted and unadapted condition. Figure 1d shows the receptive field (RF) positions and sizes for the array recordings (RF: $X = +4^{\circ}$ - 5° , $Y = -4^{\circ}$, size = ca. $4^{\circ} \times 3^{\circ}$). Taken together these figures indicate that at least some receptive fields in the "adapt" condition did span the two colour stimuli (red saccade landing points 2-3° from the dividing line). Furthermore, some of the lower saccade starting points in the "unadapted" condition would have receptive fields reaching easily into the colour zone underneath during, so the responses would be for one colour stimulus against another.

It would be helpful to plot the receptive field positions not just the saccade landing points and show that for all included data the receptive fields are fully inside the relevant stimulus boundaries.

Perhaps we were unclear, but the start and end points of the traces shown in Figure 1e do not denote the point of gaze of the animal. Instead, they mark the mean center of mass of the receptive fields (RFs), and the traces show how the center of RFs moves across the screen during saccades. Additionally, we included fixations in the analysis only if the RFs were confined within a quadrant (to avoid RF stimulation with more than one stimulus). Neurons whose RFs straddled boundaries between stimuli were not included in the analysis. This is shown in the control analysis in Supplementary Figure 2f which shows that the most extreme points of the RFs (not the center) were at a mean distance > 0 deg from the respective boundaries. Thus, the fixations included in our analysis do not span multiple color stimuli. We have now included this information in the Methods section to further clarify how trials were selected during free viewing (lines 440-442).

2.2) The authors control for microsaccade sizes but not endpoint directions and mean direction of microsaccades that could maybe impact this central results.

To address reviewer’s concern, we calculated the directions of microsaccades detected during fixations for control and adaptation conditions in each session. We then tested whether there is a preferred saccade direction (unimodal distribution) for either distribution by applying Rayleigh’s test (CircStat Toolbox, Berens 2009). We found no significant direction in the distribution of microsaccades (Figure R1) for either condition (Rayleigh’s test, $P > 0.05$). Although we found a bias towards cardinal directions in both cases, we did not find a significant difference ($P > 0.05$) in the medians of the distributions when we performed a non-parametric two sample test for circular distributions (Fisher NI, 1995). This shows that changes in tuning and decoder accuracy cannot be attributed to a difference in microsaccade direction during fixations. We have now referred to this control analysis in the main manuscript (lines 167 and 438) and included this plot in Supplementary Figure 3e.

Figure R1. Circular distribution of microsaccade directions detected during fixations, across all free-viewing sessions in adapted (pink) and unadapted (light blue) trials. The numbers along the circumference of the circle represent degrees and the numbers along the radial direction denote probability values for the distribution.

2.3) Figure 1g shows that the bulk of the recorded neurons did not respond well to the tested stimuli (<10 spikes/s). Thus, it is unclear what contribution, if any, a large number of the analysed neurons would make in a related perceptual task. Also, low firing rates can lead to spuriously high or highly variable interneuronal correlations. There should be a reasonable inclusion criterion for analysing the data further (e.g. ≥ 10 spikes/s).

We would like to clarify that, first, we have already implemented a firing rate threshold of 5 spikes/s, i.e., we excluded from the analysis the neurons with low firing rates. This threshold was carefully chosen based on prior visual cortex studies conducted in multiple labs, including our own (Gutnisky and Dragoi 2008, *Nature*; Cohen & Maunsell 2009, *Nature Neurosci*; Mitchell et. al., 2009 *Neuron*; Cohen and Kohn 2011, *Nature Neurosci*, Beaman et. al., 2017, *Nature Comm*; Shahidi et. al., 2019, *Nature Neurosci*). Second, we implemented an additional threshold to eliminate the effect of outlier firing rates on noise correlation values. We removed trials whereby the absolute value of the z-scored firing rate of either neuron was > 3 SDs (Zohary et. al., 1994, *Nature*; Smith and Kohn 2007, *J Neurosci*). The implementation of both these thresholds prevents the contamination of correlation values due to spurious firing rates. Third, based on reviewer’s suggestion we increased the threshold further to only include neurons with firing rates > 10 spks/s, and hence observed that pairwise noise correlations exhibited the same trend, i.e., adaptation reduced the variance in the distribution (two-sample F-test, $P < 0.0001$) of noise correlations without affecting the mean (Figure R2; but see our response to comment # 3). Finally, it has been shown that the magnitude of noise correlations increases with the mean firing rates of neurons (de la Rocha et. al., 2007, *Nature*; their low firing rates were < 5 Hz, and mean correlations were < 0.1). Thus, low firing rates are more likely to generate low

Figure R2. Distribution of pairwise noise correlations during free-viewing experimental sessions for monkeys T (left) and M (right) for control (blue) and adaptation (orange) conditions.

correlations as opposed to high spurious correlations. We have now included this plot in Supplementary figure 2a.

2.4 Also, since in the first experiment, reward was given at regular intervals unrelated to task stage, this could have affected neuronal responses as well.

To address this issue we examined the frequency of reward delivery in control versus adaptation trials. We subsampled a subset of trials from each condition and calculated the number of times reward was dispensed in each trial. We performed this sub-sampling 100 times and calculated reward frequency in each condition for 12 out of 16 sessions (Figure R3). There was no significant difference in reward frequency across conditions (Wilcoxon signed rank test, $P > 0.1$). This analysis shows that adaptation-induced changes in neuronal tuning cannot be attributed to differences in reward delivery across conditions. We have referred to this control analysis in lines 168-170.

Figure R3. Reward frequency in unadapt and adapt conditions per trial. Open circles and error bars represent mean value and s.e.m for individual sessions, respectively.

3) The authors indicate changes in noise correlations distributions across neuronal population (Fig. 1h) in the freely viewing condition, which are interesting, but were unfortunately this was not confirmed in the more controlled passively viewing condition. This suggests they are not a general mechanism for cross-stimulus adaptation.

The reviewer is correct, the reduction in the variance of correlations was not consistent across experimental paradigms, although we did observe some level of consistency in that the mean value of correlations remained unchanged and decoder accuracy increased after adaptation. However, we addressed this important concern with additional data analysis examining the explained variance in the population response before and after adaptation. This is motivated by the fact that our population decoder considers the shared variability not merely between pairs, but for the entire population of neurons. Therefore, in the revised manuscript we have reanalyzed our data by performing principal component analysis (PCA) on a trial-by-trial basis for the entire population of cells (similar analyses as those in Ni et. al., 2018, *Science*, and Valente et. al., 2021, *Nature Neurosci*), in both unadapt and adapt conditions for each experimental paradigm. We hypothesized that decoder performance in the adapted condition is improved because the population response is more decorrelated, i.e., the shared variability between neurons is reduced relative to control. Principal Component analysis was performed on responses in each trial (e.g., firing rates during fixations) of all recorded neurons in the control condition, i.e., PCA was done on a $M \times N$ matrix where M denotes the trials and N the number of neurons. To control for the effect of high variance in single neurons (outliers) dominating the PCA, we removed neurons whose variance exceeded 2 std above the mean variance of all the neurons in a session. We then projected the trial-by-trial activity of the same neurons during the adaptation condition onto the PC axis obtained for the control condition and examined the ratio of the variance explained by each principal component in control and adaptation (Figure R4A, B) by focusing on the mean ratio for those PCs that captured 90% of the variance of the population response in the control condition. Interestingly, we found that there was a significant reduction in variance (2-sided

Wilcoxon signed rank test) explained by the PCs post adaptation in the vast majority of sessions across both free-viewing and passive fixation experiments (Figure R4 C, D). This was also true if we analyzed the effect of adaptation on just the 1st PC, which shows that the analysis is robust to the number of PCs chosen (Figure R4 E, F). That is, our analysis shows that in the higher dimensional space of neural responses there is a transition from an ellipsoid to a more spherical clustering of neuronal responses post adaptation indicating an overall decorrelation of the population response, which could be the likely mechanism for the improvement in decoder accuracy in both free-viewing and passive fixation conditions. Given these new, exciting results, we replaced the pairwise correlation analysis in Figure 1, 3 and 4 with the PCA analysis explained above, and included a detailed description of how the analysis was performed in the Methods section (lines 505-518). The PCA analysis involving just the 1st PC has also been included as Supplementary Figures 2b and 9 a,b. Finally, the pairwise correlation results for free viewing have been moved to Supplementary Figure 2a, and that for the passive fixation experiments has been moved to Supplementary Figure 8.

Figure R4. (A, B) Ratio of variance captured along PC axes for control and adaptation conditions in an example free-viewing and passive fixation session. Shaded gray patch indicates the number of PCs that account for 90% of the variance in the control condition. Filled gray circles denote ratios evaluated along each PC axis. Black solid lines show exponential fit to aid visualization (C) Ratio values for individual free-viewing sessions recorded in both monkeys. (D) PC ratio values for all passive fixation sessions across 3 monkeys in different cross-feature adaptation cases. (E, F) Ratio of the 1st PC across both experimental paradigms. Light shaded circles represent sessions where the ratio was not significantly greater than 1.

4.1) Another interesting feature in the data are the neurons that show tuning in the adapted state, but not in the unadapted state (Figure 2d). However, if one looks at Figures 3b and 3c, there seems also to be a sizeable number of cells that show the opposite behaviour. Given that most of the analysed cells show a low OSI or CSI, this suggests that we are dealing with a noisy data set, where by chance some cells show this behaviour. It might be helpful to restrict the analysis to cells with a clear response and tuning to the stimulus and show the effect in this more controlled data set.

We were aware of the issue mentioned by the reviewer while performing our analysis. For calculating CSI/OSI we only considered neurons that had a peak firing rate > 10 spikes/s. We then performed a Rayleigh test to assess the significance of tuning for each neuron followed by a Holm-Bonferroni correction for multiple comparisons. Hence, our results are already restricted by implementing a firing rate threshold and the appropriate statistical tests. We have now revised Figure 3a to show only the neurons with firing rates > 10 spikes/s that were originally used to generate Figure 3b.

4.2) *The authors show an increase in decoding accuracies that is about the “net gain” in populational encoding. I think it would be informative to look at the dynamics of this gain of information and changes in tuning. For example to split session trials between beginning and end of the session and compare decoding performance and tuning curves of the neurons in the adapted conditions, if the authors are correct, I would expect that during phases of the experiment, even in the adapted state*

when the tuning is stronger to have more effect on the decoder. Thus, a link between the two observations could be demonstrated.

To address reviewer’s concern, we would like to clarify a few important points regarding the experimental paradigm. First, the concern raised by the reviewer would be applicable if sessions were run in a block design whereby a set of trials are run with an adapter in every trial, followed by a set of trials without an adapter, etc. However, in the free-viewing paradigm, monkeys voluntarily looked at different points in their field of view and hence there is no inherent block structure to the trials. Furthermore, adapt and unadapt conditions were randomly interleaved across trials. Second, we examined the effect of rapid adaptation at the time scale of visual fixation (200-300 ms) and not the effects of prolonged adaptation (several seconds). We nonetheless followed reviewer’s suggestion and split the mean responses of neurons to test stimuli into early (1st half) and late (2nd half) trials for both control and adaptation trials (Figure R5), but did not find statistically significant changes (Wilcoxon’s signed rank test, $P > 0.05$) between early and late trials during control (unadapt) or adaptation conditions.

Figure R5. Scatter plots showing the mean firing rates of neurons during early and late phases of free-viewing task during unadapt (filled blue circles, left) and adapt (red filled circles, right) pooled across all 16 sessions. Each filled circle represents a single neuron.

5) *For the 2nd experiment, the authors suggest that pairs neurons change their interneuronal correlations (from positive to negative and vice versa) between the unadapted and adapted state (Figure 3h). Unfortunately, the combined distribution does not look bimodal, but rather Gaussian. Since adapted and unadapted trials have been collected in an interleaved fashion for each cell pair, it seems possible that this result appears randomly by dividing the trials of the cell pairs and then looking at the extremes of what is a single distribution. A random permutation test on could clarify this questions.*

To address reviewer’s concern, we performed random permutation testing (see also our response to concern #3 where we explain why the pairwise correlation analysis was moved to Supplemental Information). Specifically, we randomly sub-sampled pairs from the entire population irrespective of whether pairwise correlations were positive or negative. We performed this sub-sampling 10000 times by drawing the same number of positive and negative correlated pairs as

Figure R6. (A, B) Distribution of the mean change in pairwise noise correlations for randomly sampled pairs (10000 times). Arrows represent mean value of changes for positive and negative correlations for actual pairs without random sub-sampling.

present in the actual data. We calculated the change in correlations for these randomly sampled pairs and plotted the distribution of the mean of the differences (Figure R6 A, B). The mean change in correlations for both negative and positively correlated pairs were significantly different from the means obtained from random sub-sampling (2 tailed Wilcoxon signed rank test, p val < 0.0001). This shows that the changes observed in noise correlations depend on their sign in the unadapted condition, and that these changes cannot be explained by what would be expected by chance.

6) *There are neurons in V1 that have a range of orientation and colour selectivity and their combination (Johnson, Hawken, Shapley J Neurosci 2008; Garg, Li, Rashid, Callaway, Science 2019; cited by the authors). And V1 neurons have been shown to have different dynamics on different timescales for specific and non-specific inputs from just outside the classical RF, which is also stimulated here (Henry, Jazayeri, Shapley, Hawken elife 2020). Given that in the 2nd experiment adaption show the expected decline in firing rate with adaption (figures 3i and 4e), it seems likely that the V4 responses could be explained by inheritance of signal changes from V1 from neurons that show joint encoding of orientation and colour. A distinct clear model of the underlying neural mechanisms V4 is not explicitly offered. The manuscript would benefit from theoretical underpinning and the proposal of a mechanistic model for the cross-feature adaptation.*

We appreciate reviewer's suggestion urging us to explore a mechanistic model for explaining the findings reported in our study. Although a detailed full-scale model explaining our results may be beyond the scope of our study, we have now included a figure in the main manuscript (Figure 5) and associated text in the Discussion section describing possible mechanisms through which cross-feature adaptation could manifest in visual cortex (lines 289-311). Briefly, we propose that models relying on neurons encoding independent/non-overlapping features to provide input to V4 are inconsistent with our findings, and that cross-feature adaptation effects could arise due to the mixed feature selectivity/specificity of inputs to area V4. We would also like to mention that based on recent studies (cited by us), it is possible that cross-feature adaptations could arise earlier in the visual pathway, e.g., V1, and these effects could carry over to downstream visual areas. However, we do not imply that area V4 is the source of cross-feature adaptation or that it represents the first cortical site where the effects we report are observed. Additionally, to the best of our knowledge, cross-feature adaptation effects have never been investigated in visual cortex or any other sensory cortical area.

Minor points:

7) *It would be helpful to the reviewer to have page and /or line numbers.*

As per reviewer's request, we have now included page and line numbers in the manuscript.

8) *The main results should be confirmed separately for each monkey in the supplement.*

We have now included plots in the actual manuscript summarizing the main results for each monkey separately (Fig. 1i, k and Fig. 3f, h) as well in the supplementary material (Fig. 2a; Fig. 5, and Fig. 9).

Reviewer #2 (Remarks to the Author):

Primary comments

Truly cross-feature? Much of the significance of the study hangs on what exactly is cross-feature. I feel this needs to be more deeply examined. That is, stimuli can be characterized as in a ‘color/surface’ category or an ‘orientation/achromatic’ category. However, each cell’s preference in V4 may not be so simple. Whether each cell’s response deviates from its ‘feature category’ then depends on some quantitative characterization of its multi-feature tuning space. Does one really need to invoke the cross-feature characterization or will any salient change lead these observed changes?

We appreciate reviewer’s thoughtful comments regarding multi-feature representation in visual neurons, especially in area V4, which is known to represent a diversity of features. We would like to clarify that we refer to ‘cross-feature’ from a stimulus, not cell-selectivity, standpoint. As the reviewer acknowledges, stimuli can be characterized as in a ‘color/surface’ category or an ‘orientation/achromatic’ category, which can be considered orthogonal features/categories. Individual neurons, on the other hand, could have mixed stimulus selectivities, or could exclusively encode orientation or color. Our work adds to the large body of literature regarding the anatomical separability of color and orientation representations by presenting new evidence that adapting to a feature that neurons are untuned to (e.g., orientation) often influences the tuning and correlated variability in responses to an orthogonal feature (e.g., color). This is only possible if the inputs that V4 neurons receive have mixed feature selectivity. Our findings may generalize to other features, namely shape (2D, 3D), texture, etc., if these inputs are combined in specific ways.

Cell types? What is unclear to me is, for the cells that reveal enhanced cross-feature tuning, are these cells hue broadband and hue tuned cells that become enhanced in their preference, or are they oriented cells that do not typically have color tuned response? I think the meaning of this enhancement would be quite different depending on whether this is observed in e.g. a color domain or an orientation domain. That is, every cell can be viewed as a multi-dimensional entity with some bias towards a baseline state, one which is predominantly biased towards ‘orientation’ in orientation domains and toward ‘color’ in color domains. Evoking color tuning to a color broadband cell may not be cross-feature, while evoking this in an orientation cell would qualify as cross-feature. Note that this is an oversimplification, as there are many cells in V4 of great complexity. Did the authors try to classify, in their paradigm, whether cells are color selective/orientation non-selective, orientation selective/color non-selective, color and orientation selective,...?

This is an interesting question regarding the tuning of neurons to both features prior to adaptation. We have examined the tuning preference of neurons that exhibited adaptive changes to both color and orientation stimuli. Specifically, 63.7% of the neurons were only tuned to color, 13.2% were only tuned to orientation, 8.8% were tuned to both color and orientation, and 13.2% were not significantly tuned to either color or orientation. We have included these tuning statistics in the Results section (lines 189-192). Note that even cells that were not tuned yet responsive to the adapting stimulus can still exhibit changes in tuning post adaptation. Further, we used the term ‘cross-feature’ to refer to rapid adaptation of neurons when exposed to one feature that subsequently leads to changes in response properties for a different feature. For instance, based on the findings of prior studies, one would not expect color tuning of neurons to change unless color stimuli are used as adapters (same feature). We show that this is not the case and there are cross-feature interactions. We hypothesize that these effects could extend beyond color and orientation to other features that V4 neurons are known to represent (shapes) as long as the mixed feature signals from upstream areas are transmitted via feedforward channels.

Discussion weak: There is a lack of discussion regarding what mechanisms/circuits might underlie their observations. This cross-feature interaction is reminiscent of cross-orientation effects whereby an oriented stimulus enhances the cells of the preferred orientation and relatively suppresses those of

orthogonally oriented cells. This push-pull effect effectively sharpens orientation tuning and 'tightens' the circuit. Do the authors think their findings might be explained by something analogous?

We agree with the reviewer, and now discuss a mechanistic model that could potentially explain the findings reported in our study. This is described in Figure 5 and the relevant text in the Discussion section that suggest possible mechanisms through which cross-feature adaptation might operate in visual cortex (lines 289-311). Briefly, we propose that independent/non-overlapping feature inputs is inconsistent with our findings and that cross-feature adaptation effects arise due to mixed feature selectivity/specificity of inputs to V4 neurons.

What do the longer latencies mean? Why the repulsive shift?

We believe that longer latencies indicate the effect of the adapter and possibly recurrent processing within area V4 caused by the activation of the adapter (this was moved to Supplementary figure 6). To address the comment regarding the change in preferred color (repulsive shift), in the revised version we re-plotted the actual value instead of the absolute value of the change in preferred color due to orientation adaptation, and hence observed a range of both positive and negative changes in preferred color. At the population level (mean of the distribution across all neurons) we observed a small mean change of 1.4°. However, in contrast to inter-feature adaptation, these changes cannot be described as repulsive or attractive as the adapting feature (orientation) is different from the test feature (not color). Additionally, we observed substantial changes (>22.5° or < 22.5°) in only 18% of the neurons. These findings have now been included in Supplementary figure 7.

Minor comments

I did not find Fig 3e, 3f compelling.

We have moved Fig 3f to the supplementary materials section.

Reviewer #3 (Remarks to the Author):

Main comments:

1. Why is cross-feature adaptation important and why is it useful? In general, adaptation is useful because it adjusts the sensitivity of neurons in response to the intensity of external stimuli. Why does the nervous system want to adjust its sensitivity between features, especially after saccades? Is there any theoretical or psychophysical literature on this point?

Our main focus in the manuscript was to examine adaptive coding at the single cell and population level that may resemble rapid adaptation occurring during the visual exploration of natural scenes. Although we agree with the reviewer that adaptation enables neurons to adjust their responses to the changing intensity of stimuli, adaptation can be more generally viewed as a dynamic phenomenon whereby single cells and networks change their encoding properties after exposure to a stimulus of fixed structure. For example, work from our lab and others has shown that rapid adaptation to orientation and color stimuli leads to changes in neurons' selectivity and tuning strength, correlated activity, and population coding at time scales of seconds to hundreds of ms (during visual fixation), e.g., Dragoi et. al., 2000, *Neuron*; Dragoi et. al., 2002, *Nature Neurosci*; Yao and Dan, 2001, *Neuron*; Kohn and Movshon, 2003, *Neuron*; Gutnisky and Dragoi, 2008, *Nature*; Hansen and Dragoi, 2011, *PNAS*; Nigam et. al., 2021, *Science Advances*. It is important to note that in all previous experiments the adapter had the same mean intensity as the test stimuli to prevent an adjustment of neuronal responses to changing intensity. As pointed out in Figure 1a, natural scenes are rich in diverse features, especially color and orientation. Thus, during scene exploration through saccadic

eye movements, receptive fields of neurons are likely to be stimulated by different features. Despite its ubiquity during natural viewing, cross-feature adaptation has been largely overlooked based on prior findings that different features are processed independently by non-overlapping sets of neurons. However, recent work in V1 has shown that the same neurons can represent multiple features (color and orientation, Garg et. al., *Science*, 2019). Our findings demonstrate the effects of such multi-feature representation in visual cortex in the context of natural viewing and indicates how visual cortex might have evolved to optimize the extraction of information from complex natural scenes containing multiple features. We strongly believe that our findings may lay the foundation for further exploration of rapid plasticity that occur during viewing natural scenes. Additionally, our work raises the question of how simultaneous multi-feature adaptation during viewing of natural scenes influences stimulus coding.

2. The results report only on cross-feature adaptation. This greatly hampers the interpretation of the results. There is a need to compare inter-feature adaptation with cross-feature adaptation. For example, what are the differences in color tuning and decoder performance between color adaptation and orientation adaptation?

We believe the reviewer is referring to whether there are differences between inter and intra-feature adaptation. To address this concern, we analyzed additional sessions for both color and orientation stimuli (6 sessions for each feature; we analyzed color and orientation adaptation in 117 and 77 neurons respectively), not included in this manuscript. However, we observed qualitatively similar results to those for cross feature adaptation. There was a significant increase in CSI/OSI post adaptation in case of either feature (Figure R7 A-B, D-E, $\Delta\text{CSI}_{\text{color}}(\text{adapt-unadapt}) = 42.1 \pm 2.1\%$, $P < 0.001$; $\Delta\text{CSI}_{\text{ori}}(\text{adapt-unadapt}) = 41.1 \pm 1.3\%$; $P = 0.01$; Wilcoxon signed rank test). Inter-feature adaptation improved decoding of stimuli post adaptation (Figure R7 C, F, $\% \Delta \text{dec acc}_{\text{color}} = 22.4 \pm 4\%$, $\% \Delta \text{dec acc}_{\text{ori}} = 33.4 \pm 13.4\%$). In addition, please note that there has been extensive previous work on inter-feature adaptation conducted by our lab and others in the orientation domain (see references in response to previous comment). Furthermore, a recent publication from our lab (Nigam et. al., 2021) quantified, among other results, the effects of rapid color adaptation in V4 and revealed repulsive/attractive shifts in color tuning based on the difference between the neurons' preferred color and that of the adapter. Thus, adaptation within the same feature space exhibits similar effects irrespective of the feature (color or orientation) used to study inter-feature adaptation. The focus of our current work, however, was to venture beyond inter-feature adaptation, and examine cross-feature adaptation, which occurs during the visual exploration of natural scenes.

3.1 More explanation of the experimental protocol and analysis is needed. In particular, it is very important to know where the receptive fields of the neurons were located relative to the boundaries of

Figure R7. (A) CSI values of color tuning in unadapt and adapt conditions for color adaptation. (B) Histogram of difference in CSI values (adapt-unadapt). Arrow represents mean change in CSI and asterisks denote $P < 0.01$ (Wilcoxon signed rank task). Percentage change in decoder accuracy values (adapt - unadapt) for analyzed sessions. Each unfilled circle represents a session and error bars (s.e.m) were calculated by running the decoder multiple times with a different set of training and test trials. (D-F) Same as in A-C except for when the test and adapting stimulus were oriented gratings.

the adapting, control, and test stimuli in the free-viewing condition. The method corresponding to Figure 1e, where control and adaptation trials are sorted based on the start and end points of the saccade, only indicates that the receptive field was within the boundaries of the monitor. However, the receptive field is located to the lower right of the fixation point, and it is unclear whether the receptive field is restricted to the adapting and control stimuli in adaptive and control trials, respectively.

We would like to clarify that the start/end points of the traces shown in Figure 1e do not denote the point of gaze of the animal. Instead, they represent the mean center of mass of the receptive fields (RFs), and the traces show how the center of RFs move across the screen during saccades. Additionally, we included only those fixations for which RFs were confined within one quadrant in order to avoid RF stimulation with more than one stimulus (neurons whose RFs straddled boundaries between stimuli were not included in the analysis). This is illustrated for the control analysis in Supplementary Figure 3f showing that the most extreme points of the RFs (not the center) were at a mean distance greater than 0 deg from the respective boundaries. We have now included this information in the Methods section to further clarify how trials were selected in the free-viewing experiments (lines 431-442).

3.2 Furthermore, can the direction of the saccade affect the neuron's response.

To examine this issue during the free viewing experiments, we compared neurons' firing rates during fixations on test stimuli 1 and 2. These fixations are preceded by different saccade directions due to the position of test stimuli 1 and 2 in two different quadrants (Fig 1e main manuscript). We performed this analysis separately in both unadapt and adapt conditions. If saccade direction influences firing rates, they would be consistently higher for all neurons associated with a particular saccade direction. However, we found no significant difference in firing rates across neurons in both monkeys (N = 159 neurons) in either unadapt or adapt trials (Wilcoxon signed rank test, $P > 0.05$). This is consistent with Fig 1g in the main manuscript where we reported no significant differences in firing rates averaged over the 2 test stimuli across both conditions. We have included this control analysis in the Results section (lines 168-170).

4. The discussion section is poor and needs to be more refined.

We have now expanded the discussion section to include possible mechanisms through which cross-feature adaptation might operate in visual cortex (Fig. 5) and how a decrease in population wide shared variability during cross feature adaptation leads to an improvement in stimulus discrimination (lines 289-311 and lines 323-337).

4a. Why does the decoder perform better in adaptive trials versus non-adaptive trials? The mechanism needs to be discussed. Can it be explained by changes in noise correlation? In general, decoding performance depends on the average noise correlation and not on the distribution.

4b. The results on noise correlation seem to be different between the free-viewing experiment and the fixation experiment. How can this difference be explained?

4c. What neural circuitry is responsible for the increase in the negatively correlated pairs and the decrease in the positively correlated pairs? What is the significance of this change in noise correlation?

We appreciate reviewer's suggestion to discuss a mechanistic understanding of how cross feature adaptation improves decoder accuracy. To address this concern, we employed a more general approach for quantifying correlated activity in neuronal populations. This was motivated by the fact that the neural decoder takes in account the shared variability not merely between pairs but the entire population of neurons. Thus, in the revised manuscript we re-analyzed our data by performing principal component analysis on the trial-by-

trial responses of the entire populations of neurons (e.g., Ni et. al., 2018, *Science*; Valente et. al., 2021, *Nature Neurosci*) in unadapt and adapt conditions for both experimental paradigms.

To implement this, we performed a Principal Component analysis on trial-by-trial responses (firing rates during fixations) of all recorded neurons in the control condition, i.e., PCA was done on a $M \times N$ matrix where M denotes the trials and N the number of neurons. We then projected the trial-by-trial activity of the same neurons during the adaptation condition on to the PC axis obtained for the control condition. Next, we calculated the ratio of the variance explained by each principal component in the control and adaptation condition (Figure R4A, B). We took the mean value of the ratio for PCs that captured 90% of the variance of the data in the control condition. Interestingly, we found that there was a significant reduction in variance (2-sided Wilcoxon signed rank test) explained by the PCs post adaptation in a majority of sessions across both free-viewing and passive fixation experiments (Figure R4 C, D). This was also true if we analyzed the effect of adaptation on just the 1st PC which shows that the analysis is robust to the number of PCs chosen (Figure R4 E, F). This analysis shows that in the higher dimensional space of neural responses there is transition from an ellipsoid to a more spherical clustering of responses post adaptation indicating an overall decorrelation. This could be a likely mechanism for the improvement in decoder accuracy in both free and passive fixation experimental paradigms reported in this study. We have replaced the pairwise correlation analysis in Figure 1, 3 and 4 with the PCA analysis explained above and included a detailed description of how the analysis is performed in the Methods section (lines 505-518). The PCA analysis involving just the 1st PC has now been included as Supplementary Figures 2b and 9 a,b. Finally, the pairwise correlation results for free viewing have been moved to Supplementary Figure 2a, and passive fixation experiments to Supplementary Figure 8.

Minor comments

1. The title is too general. The paper deals with cross-feature adaptation, and the experiment does not deal with naturalistic signals themselves.

We used the term “naturalistic signals” to refer to neuronal responses generated during unrestrained visual exploration mimicking those generated during viewing of natural scenes. In order to address the concern of the reviewer we have now chosen a title that most accurately describes our study. ***Adaptive coding across visual features during free-viewing and fixation conditions.***

2. Which hemisphere did you record from in each monkey? Is it the left hemisphere in all monkeys?

We recorded from the left hemisphere in monkeys R, M and T. We have now included this information in the Methods section (line 359).

3. Which monkeys participated in each experiment?

Monkeys T and R participated in the free-viewing experiments. Monkeys T and M participated in the passive fixation experiments. The reviewer can find this information along with the number of sessions recorded in each animal in the results section (lines 122, 183-184 and 249-250).

4. How did you extract the "pseudo-trials" for the free-viewing task? What are the criteria?

We first detected fixations during free-viewing by thresholding eye velocity (100 deg/s, Smeets and Hooge, 2003, *JNeurophysiol*) (see Fig. 1f). Once fixations were detected, pseudo trials (control and adaptation) were determined based on the sequence of consecutive fixations on the quadrants. If fixation on one of the test stimuli was preceded by a fixation on the gray quadrant, then the fixation on the test stimuli was labelled

as a ‘control’ trial. However, if fixation on test stimuli was preceded by fixation on an adapting stimulus, then the fixation was labelled as an ‘adaptation’ trial. We have now included additional text in the Methods section (lines 431-442) that further clarifies how trials were classified as control and adaptation trials.

5. In the free-viewing task, how was the time window for calculating spike counts for decoder analysis set?

Spike counts were calculated during the duration of the fixation on each of the quadrants. Mean fixation durations were 380 ± 5 ms and 385 ± 7 ms in unadapted and adapted trials respectively. We performed control analysis as shown in Supplementary Figure 2a, b to show that fixation durations on test stimuli or adapter were not significantly different between control and adaptation trials. Please also see the caption for Supplementary Figure 3a, b.

6. Figure 1d. Why is the origin not the fixation point?

Receptive fields were mapped in a separate set of passive fixation experiments. The fixation point had to be moved towards the upper right portion of the monitor to bring the RFs of the neurons within the spatial extent of the monitor.

7. Figure 1j. Why is the decoder performance for unadapted stimuli barely above chance level? It seems inconsistent with the fact that many V4 neurons are selective for color and orientation.

V4 neurons are selective for color and orientation, but the actual difference between the 2 test stimuli is quite small (color stimuli separated by 22.5° in hue space and grating stimuli separated by 11.25° in orientation space). However, as shown in Figure 1i, 6 out of 8 sessions (ori adapt, color test) and 4 out of 8 sessions (color adapt, ori test) show decoder accuracies above chance levels in the unadapted condition. In general, we observe stronger tuning to color stimuli compared to orientation stimuli across sessions which would explain the lower control decoder accuracies for orientation compared to color. To understand why we observed lower decoder accuracy values in the control condition for either feature, one has to consider the following two factors: First, we selected pairs of color and orientation test stimuli in the free-viewing experiments based on maximizing the population firing rate, and not necessarily that of individual neurons. An important thing to note is that neurons exhibit heterogeneity in their preference to color and orientation and hence the population tuning curve has a broader structure compared to tuning curves of individual neuron. Second, we performed decoder analysis on the entire set of stimulus responsive neurons (tuned and untuned) and not restricted to only significantly tuned neurons. The combination of both these factors likely gave rise to decoder accuracies close to chance levels for the pair of color/orientation stimuli chosen in the free-viewing experiments during unadapted trials.

8. Figure 1j. Where are the open circles? How many sessions were conducted on each monkey?

We have modified Figure 1i to include both filled and open circles to denote sessions recorded from monkeys R and T. A total of 8 sessions were recorded from each monkey and were evenly split between 2 conditions: color adapter, orientation test and vice versa.

9. Figure 3. The title should be adjusted as in figure 4.

We have modified the contents of Figures 3 and 4. Please see the new modified figures.

10. Figure 3b. What are the dotted lines at 0.1? If they do not represent statistical criterion, they should be omitted. Make the inserted figure larger.

The dotted lines at 0.1 were included for visualization purposes, but as per reviewer's request we have now removed the dotted lines in both panels (Figure 3b and 4c). Additionally, we have increased the size of all insets associated with the figures in addition to Figure 3c.

11. Figure 3j. Is the data from one monkey? If the data are from different monkeys, use different symbols.

We have modified Figure 3f (note changes made to Figure 3) to include filled and open circles to denote sessions recorded from monkeys T (10 sessions) and M (7 sessions) respectively.

12. Figure 4d: Why is the format different from Figure 3h?

Note we have removed Figure 4d from our manuscript as we included new analysis to quantify population shared variability instead of pairwise statistics. Please see response to main comment # 4.

13. Figure 4f. Which monkey is this data from? Why is the decoder performance for unadapted stimuli lower than zero?

The data shown in Figure 4d (note changes made to this figure) was recorded in monkey T. We would like to clarify that the raw values of decoder performance is **not** less than 0. The decoder performance shown in Figure 4f was calculated by taking the difference of the actual decoder performance and that expected by chance (50%), and hence it represents the decoder performance above chance. We have now included additional text in the Methods section (line # 528-531) to better explain our analysis.

14. I don't think Supplementary Figure 1 is necessary. This has already been pointed out in Fig. 1a.

We included additional examples of natural scenes containing a mix of orientation and color stimuli to emphasize that such spatial distribution of features occur pretty frequently in natural scenes and is not specific to one instance of a natural image used in Figure 1a.

15. The legend in Supplementary Figure 2b is actually the legend for Supplementary Figure 2c, and the legend for Supplementary Figure 2b is missing.

We thank the reviewer for pointing out this error. We have now fixed this issue in the captions accompanying Supplementary Figure 3 (previously supplementary figure 2).

REVIEWER COMMENTS

Reviewer #2 (Remarks to the Author):

This study addresses an important question regarding neuronal adaptation during natural viewing conditions. That is, during natural eye movements, the statistics of each viewed patch may differ significantly, entailing stimulation by features of different feature spaces (e.g. orientation vs color content). However, few studies have systematically examined the effects of cross-feature adaption on neuronal response.

The authors present a novel behavioral paradigm that simulates changing feature content experienced during natural looking conditions; as well, they have taken the effort to relate these natural fixation conditions to more controlled fixation conditions. The finding that feature-specific adaptation enhances cross-feature tuning is a novel and important finding; the finding that this occurs via decorrelation in population response (decreased shared variability) resonates with many other studies invoking the merits of population decorrelation (e.g. J.Reynolds, M. Cohen). These data fit well with and extend the view that visual neurons are multidimensional entities with a baseline state (e.g. orientation-selective, color-selective) that can be dynamically tweaked based on spatial and temporal context.

In the previous review, I raised concerns regarding the meaning of 'cross-feature', the behavior of different cell types, and weakness of discussion. In the current response, the authors have reasonably addressed these concerns. This revised version is significantly improved in clarity. Explanation of decoder performance and determination of PC variance is clearer. Added description of controls (e.g. line 158 paragraph) also increase confidence in results. The inclusion of "63.7% of the neurons were only tuned to color, 13.2% were only tuned to orientation, 8.8% were tuned to both color and orientation, and 13.2% were not significantly tuned to either color or orientation" is also critically important, as this provides an understanding of the recorded population distribution. The study is carefully conducted and statistical analyses solid. The following comments are meant to further improve the discussion.

(1) The discussion could still benefit from inclusion of additional considerations which I provide for the authors to consider. (a) Analogous to observed cross-orientation inhibition (e.g. in V1 Hu et al 2020 Cerebral Cortex 30(10):5532) and cross-sensory inhibition (e.g. Mehta and Schroeder 2000 Cereb Cortex 10:343), the current results suggest there are cross-modal (color vs orientation) effects. That is, seeing an oriented stimulus preps the 'color' system by increasing variance across the population, and potentially across a range of different hues, providing a means by which the cortex is ready for any hue that is encountered in the following glance. Similarly, seeing an even field of color preps the orientation system by increasing variance across different orientations, enabling rapid emergence of orientation tuned responses. From a functional organization point of view, this would imply (a) a push-pull between color and orientation domains in V4, and (b) an increased variance and sharpened tuning across hue

domains in the color domains (orientation domains) under orientation (color) adaptation conditions (e.g. Tanigawa et al 2010, Li et al J Neurosci 4:202, Du et al 2022 Frontiers in NS).

(2) There is one important aspect that is not discussed. Psychophysically, does viewing one feature (e.g. orientation) enhance the discriminability of another feature (e.g. hue)? Please add citations.

Other questions:

(3) Functional organization. Given that roughly 2/3 of the neurons were tuned to color and 20% to orientation, the Utah array may have been located more in color associated regions and less so in orientation regions. Was there any obvious spatial clustering of the 'color' vs 'orientation' preferring electrodes in V4? Was the distribution of response preferences similar for the two monkeys? I.e. were the Utah arrays implanted in potentially different 'bands' of V4?

(4) Line 203 states that "A significant proportion of color untuned neurons (18%) became significantly tuned". Does this mean 18% of the entire population or 18% of the color untuned neurons (13.2+13.2%), i.e. roughly 7% of the total neuron population, became significantly tuned to color.

(5) Does the 8.8% color&orientation tuned population (presumably the 'color-oriented neurons' e.g. Shapley 2005 J Neurosci 28:8096, Engel 2005 Neuron 2005 45:613) exhibit significant cross-feature tuning enhancement? The meaning of increase in tuning for these neurons may differ significantly from neurons which shift from untuned to tuned.

Reviewer #3 (Remarks to the Author):

The authors have addressed some of my initial concerns, but there are still many problems with this paper. The numbers below are my initial comments.

Main comments

1. Why is cross-feature adaptation important and why is it useful? In general, adaptation is useful because it adjusts the sensitivity of neurons in response to the intensity of external stimuli. Why does the nervous system want to adjust its sensitivity between features, especially after saccades? Is there any theoretical or psychophysical literature on this point?

The authors have not addressed this issue. The literature they give is about rapid adaptation within feature that are already known to be important, and does not address the importance of adaptation across features. It is important that the authors explain why cross-feature adaptation is important and why is it useful. Without an adequate explanation, I cannot assess the significance of this paper.

2. The results report only on cross-feature adaptation. This greatly hampers the interpretation of the results. There is a need to compare inter-feature adaptation with cross-feature adaptation. For example, what are the differences in color tuning and decoder performance between color adaptation and orientation adaptation?

I appreciate the addition of the related analysis; it is interesting that cross-feature adaptation seems to have the same effect as intra-feature adaptation. Figure R7 should be presented as a supplementary figure. There are two minor points that I would like to point out. First, the X axis in Figure R7e should be OSI, not CSI. Second, why are some of the points in Figure R7c and Figure F7f above 50% (since the chance level is 50%, I would have thought the difference between adapt and unadapting would be less than 50%)?

4. The discussion section is poor and needs to be more refined.

The authors have added a new Figure 5 to illustrate a possible mechanism for cross-feature adaptation. While I agree that this particular circuit explains cross-feature adaptation, it is inadequate given the generality of this circuit. For example, if the synapses of the V2 input encoding vertical gratings are stronger, we would expect color tuning to be weaker after adaptation. In this case, the sharpening of color tuning after cross-feature adaptation of the entire population would be negated. Overall, I don't think the discussion section is refined enough.

Minor comments

13. Figure 4f. Which monkey is this data from? Why is the decoder performance for unadapted stimuli lower than zero?

I still don't understand why the decoder performance for unadapted stimuli was less than zero? I understand that the raw decoder performance is not less than zero. But why is decoder performance below chance level? This seems to be true for almost all experiments.

Reviewer #2

1. The discussion could still benefit from inclusion of additional considerations which I provide for the authors to consider. (a) Analogous to observed cross-orientation inhibition (e.g. in VI Hu et al 2020 Cerebral Cortex 30(10):5532) and cross-sensory inhibition (e.g. Mehta and Schroeder 2000 Cereb Cortex 10:343), the current results suggest there are cross-modal (color vs orientation) effects. That is, seeing an oriented stimulus preps the ‘color’ system by increasing variance across the population, and potentially across a range of different hues, providing a means by which the cortex is ready for any hue that is encountered in the following glance. Similarly, seeing an even field of color preps the orientation system by increasing variance across different orientations, enabling rapid emergence of orientation tuned responses. From a functional organization point of view, this would imply (a) a push-pull between color and orientation domains in V4, and (b) an increased variance and sharpened tuning across hue domains in the color domains (orientation domains) under orientation (color) adaptation conditions (e.g. Tanigawa et al 2010, Li et al J Neurosci 4:202, Du et al 2022 Frontiers in NS).

We thank the reviewer for sharing this insight into the functional implications of cross-feature adaptation during viewing of complex stimuli. We have now included additional text in the Discussion section (lines 375-378) addressing the above points, and also cited relevant work (reference # 29, 30 and 31) as mentioned by the reviewer.

2. There is one important aspect that is not discussed. Psychophysically, does viewing one feature (e.g. orientation) enhance the discriminability of another feature (e.g. hue)? Please add citations.

We performed an extensive literature search of psychophysical studies to verify whether cross-feature adaptation paradigms involving color and orientation have been used in the past. Surprisingly, there is a complete lack of studies on either rapid or prolonged adaptation in the cross-feature domain with respect to improved stimulus discriminability. Psychophysical evidence of cross-feature adaptation, although of a very different flavor than presented in our study, has been observed in the form of the McCullough effect (McCullough, 1965, *Science*). Although this color aftereffect generated due to long-term adaptation (2-4 minutes) implies interaction between features, it does not indicate an improvement (at any time scale) in the discriminability of features as found in our study at the single cell and population level. Furthermore, a recent study (Ghodrati et. al., 2019, *Nature Commun*) examined the effects of adaptation (~ 5 s) to contrast and luminosity on the encoding of oriented stimuli to report an improvement in discrimination thresholds post adaptation. However, unlike color/orientation encoding which is believed to be represented by anatomically distinct modules, features such as contrast and luminance are typically represented by the same population of neurons as that encoding for orientation. Given the lack of psychophysical studies, we believe that our findings in single neurons and populations could motivate future psychophysics experiments using both synthetic and natural stimuli to directly test the perceptual effects of cross-feature adaptation. We have now included additional text in the discussion section (lines 385-400) of the manuscript to address the above points.

Other questions:

3. Functional organization. Given that roughly 2/3 of the neurons were tuned to color and 20% to orientation, the Utah array may have been located more in color associated regions and less so in orientation regions. Was there any obvious spatial clustering of the ‘color’ vs ‘orientation’ preferring electrodes in V4? Was the distribution of response preferences similar for the two monkeys? i.e. were the Utah arrays implanted in potentially different ‘bands’ of V4

To address reviewer's comment, we generated a map of the stimulus preference of neurons recorded on the electrode arrays in monkeys T and M (Figure R1). We selected two representative recording sessions whereby both color and orientation adaptation were performed on the same day. These sessions were also selected because they contained the maximum number of single units recorded across all experimental sessions so as to cover as much of the spatial extent of the array as possible. However, we did not observe a clear spatial separation of the color and orientation tuned neurons in any animal. Importantly, characterizing the spatial distribution of color/orientation tuned neurons tuned to both features at a finer spatial resolution may be impossible in our study due to the relatively large spacing between electrodes (400 μm) on Utah arrays.

Figure R1. Spatial distribution of color only (red), orientation only (green) and neurons tuned to both features (blue) in monkey T (left) and M (right). Each filled circle represents the position of the electrode on the Utah array. Gray circles indicate either untuned neurons or the absence of a single unit on that electrode. Spacing between electrodes is 400 μm .

4. Line 203 states that “A significant proportion of color untuned neurons (18%) became significantly tuned”. Does this mean 18% of the entire population or 18% of the color untuned neurons (13.2+13.2%), i.e. roughly 7% of the total neuron population, became significantly tuned to color.

The statement made on line 203 indicates that 18% of all the neurons shown in Figure 3b became significantly tuned post cross-feature adaptation. Neurons that were not tuned to the test stimuli (e.g., color in case of orientation adaptation, and vice versa) in both adapted and adapted trials were not included in the analysis.

5. Does the 8.8% color & orientation tuned population (presumably the ‘color-oriented neurons’ e.g. Shapley 2005 J Neurosci 28:8096, Engel 2005 Neuron 2005 45:613) exhibit significant cross-feature tuning enhancement? The meaning of increase in tuning for these neurons may differ significantly from neurons which shift from untuned to tuned.

We analyzed the tuning properties of the ‘color-oriented’ neurons in our data set for sessions where color and orientation adaptation were performed on the same day. Please note that we encountered a small number of this type of cells and our observations might not generalize due to under-sampling. For this small cell population, we observed heterogeneous effects (sharpening, broadening and loss of tuning) due to cross-feature adaptation. We observed significant tuning enhancement in tuning strength in 3 out of 8 neurons ($\Delta\text{CSI}_{\text{mean}} = 23\%$, $\Delta\text{OSI}_{\text{mean}} = 16.7\%$). Mechanistically, the increase/decrease in tuning strength will depend on the distance between the adapting stimulus (color/grating) and the preferred color or orientation of these “color-oriented” neurons. We agree that this could be an exciting direction for future research using optical imaging that would allow high spatial resolution and cell count.

Reviewer #3

Main comments

1. Why is cross-feature adaptation important and why is it useful? In general, adaptation is useful because it adjusts the sensitivity of neurons in response to the intensity of external stimuli. Why does the nervous system want to adjust its sensitivity between features, especially after saccades? Is there any theoretical or psychophysical literature on this point? The authors have not addressed this issue. The literature they give is about rapid adaptation within feature that are already known to be important, and

does not address the importance of adaptation across features. It is important that the authors explain why cross-feature adaptation is important and why is it useful. Without an adequate explanation, I cannot assess the significance of this paper.

Although we did attempt to address the functional significance of our work in the original manuscript, we might have been unclear. We have now performed additional experiments and analyses and revised this important aspect of our manuscript as follows.

a) Importance of cross-feature adaptation. The efficient coding hypothesis (Simoncelli and Olshausen, *Ann Rev Neurosci*, 2001) predicates that visual cortical responses are adapted to the statistics of natural stimuli. That is, the higher the frequency of particular stimuli encountered during natural vision, the more efficient the neural mechanisms responsible for processing those stimuli. In light of this hypothesis, we examined how likely successive fixations occurring during free viewing could land on cross-feature stimuli (considering that each fixation episode would act as rapid adaptation). To this end, we built a statistical model of visual exploration of natural scenes to examine the features encountered in consecutive fixations. Natural scenes are characterized by a diverse set of features. In our study we focused on quantifying the distribution of two elementary features that are ubiquitously present in natural scenes: color and orientation. Based on the distribution of saccade lengths and directions obtained from the freely-viewing data in two monkeys (Figure R2A, B), we used a natural image battery ($n = 940$) from the McGill calibrated color image data base (McGill Vision research) to simulate pseudo-saccades originating from a randomly chosen point for each image. We analyzed a total of 282,000 saccades across all images (300 saccades per image). The size of image patches was chosen based on mean receptive field size of our V4 neurons. For every pseudo-saccade connecting two patches of a natural image, we quantified the strength of orientation and color signals. Orientation signals were calculated by computing an orientation selectivity index (OSI) using a Sobel filter (see Methods). Color signals were quantified by analyzing the distributions of the red, green and blue channel values across all pixels within a patch (values lying between 0 to 255). A patch was considered to be ‘colored’ if it met the following requirements: a) each of the peaks of the R, G and B channel distributions were not located within a threshold Δ (set at 30) to either 0 or 255 (black or white patches). b) The difference in the peak locations for each channel were not closer than $\pm\Delta$ (to rule out gray patches). Patches were considered ‘oriented’ if there was significant tuning (Rayleigh’s test, $p_{val} < 0.05$) based on Sobel filter analysis and did not have color information based on the criteria presented above. To ensure that patches with no significant orientation tuning did not simply contain multiple orientations

Figure R2. (A) Top and bottom represent probability density function of the length (L) and direction (D) of saccades ($n = 11253$, $L_{mean} = 7.9 \pm 0.05$ deg, $D_{circular\ mean} = 1.2^\circ$) made by monkey R, pooled across all sessions during free-viewing of stimuli. (B) Same as in A except for monkey T ($n = 25250$, $L_{mean} = 7.6 \pm 0.03$ deg, $D_{circular\ mean} = -2.8^\circ$) (C) Percentage of simulated saccades connecting patches with similar (iso-feature, light blue dots) and dissimilar features (cross-feature, orange dots). Individual dots represent the mean percentage of saccades calculated for each natural image used in the analysis. Black unfilled circles and vertical bars represent mean and s.e.m. of the distributions.

pseudo-saccade connecting two patches of a natural image, we quantified the strength of orientation and color signals. Orientation signals were calculated by computing an orientation selectivity index (OSI) using a Sobel filter (see Methods). Color signals were quantified by analyzing the distributions of the red, green and blue channel values across all pixels within a patch (values lying between 0 to 255). A patch was considered to be ‘colored’ if it met the following requirements: a) each of the peaks of the R, G and B channel distributions were not located within a threshold Δ (set at 30) to either 0 or 255 (black or white patches). b) The difference in the peak locations for each channel were not closer than $\pm\Delta$ (to rule out gray patches). Patches were considered ‘oriented’ if there was significant tuning (Rayleigh’s test, $p_{val} < 0.05$) based on Sobel filter analysis and did not have color information based on the criteria presented above. To ensure that patches with no significant orientation tuning did not simply contain multiple orientations

(multiple peaks in their OSI distribution), we eliminated from the analysis those patches containing more than one peak in their OSI distribution, where peaks were defined as local maxima in the OSI distribution > 3 s.d. above the mean. For each image we analyzed 300 saccades to calculate how often saccades connected two patches with dissimilar/similar features. Strikingly, we found that a significant percentage of saccades (23%) involved stimuli with dissimilar features (color \rightarrow orientation or orientation \rightarrow color) at the start and end points (Figure R2C). This analysis indicates that during visual exploration of natural images, neurons in the visual system will likely encounter a large fraction of dissimilar cross-features during successive fixations. Since the visual system is believed to have evolved to efficiently represent the statistics of natural scenes, cross-feature adaptation during natural viewing could provide a means to achieve this goal. We have now included panels A and B of Figure R2 into Supplementary Figure 2, incorporated panel C of Figure R2 in Figure 1 in the main manuscript and updated the Results (lines 109-130) and Methods section (lines 517-539).

b) Functional benefits of cross-feature adaptation. Our results in Fig. 1e, 3f and 4d demonstrate the benefits of cross-feature adaptation. Indeed, using linear classifiers and dimensionality reduction techniques (PCA), we showed that rapid adaptation increases stimulus discriminability by decorrelating the population response. From a functional point of view this implies that exposure to an oriented stimulus *prepares* the color encoding system by decreasing variance across the population, thereby enabling a rapid emergence of color tuned responses during the following fixation (hence increasing coding accuracy). We have included additional text in the Discussion section to elaborate on this point (lines 375-378).

c) New insights. Our study is the first investigation systematically examining the effects of cross-feature adaptation using a novel free-viewing paradigm designed to simulate natural viewing conditions. In addition, we used more controlled fixation conditions to characterize in much greater detail the effect of cross-feature adaptation on neuronal tuning curves and population discrimination performance. We have shown that neurons can adapt to stimuli they are not necessarily tuned to – this is a novel idea that may lead to a major revision of the adaptation phenomenology. Indeed, untuned neurons are usually ignored in most studies of cortical function, yet they could play a significant role in the adaptive coding of visual features.

Finally, we performed an extensive literature search of psychophysical studies to verify whether cross-feature adaptation paradigms involving color and orientation have been used in the past. Surprisingly, there is a complete lack of studies on either rapid or prolonged adaptation in the cross-feature domain with respect to improved stimulus discriminability. Psychophysical evidence of cross-feature adaptation, although of a very different flavor than presented in our study, has been observed in the form of the McCullough effect (McCullough, 1965, *Science*). Although this color aftereffect generated due to long-term adaptation (2-4 minutes) implies interaction between features, it does not indicate an improvement (at any time scale) in the discriminability of features as found in our study. Furthermore, a more recent study (Ghodrati et. al., 2019, *Nature Commun*) examined the effects of adaptation (~ 5 s) to contrast and luminosity on the encoding of oriented stimuli to report an improvement in discrimination thresholds post adaptation. However, unlike color/orientation encoding which are believed to be represented by anatomically distinct modules, features such as contrast and luminance are typically represented by the same population of neurons as that encoding for orientation. Given the lack of psychophysical studies, we believe that our findings in single neurons and populations could motivate future psychophysics experiments using both synthetic and natural stimuli to directly test the perceptual effects of cross-feature adaptation. We have now included additional text in the discussion section (lines 385-400) of the manuscript to address the above points.

2. The results report only on cross-feature adaptation. This greatly hampers the interpretation of the results. There is a need to compare inter-feature adaptation with cross-feature adaptation. For example,

what are the differences in color tuning and decoder performance between color adaptation and orientation adaptation?

I appreciate the addition of the related analysis; it is interesting that cross-feature adaptation seems to have the same effect as intra-feature adaptation. Figure R7 should be presented as a supplementary figure. There are two minor points that I would like to point out. First, the X axis in Figure R7e should be OSI, not CSI. Second, why are some of the points in Figure R7c and Figure F7f above 50% (since the chance level is 50%, I would have thought the difference between adapt and unadapting would be less than 50%)?

We thank the reviewer for the initial suggestion to analyze our data on iso-feature adaptation. It complements nicely our cross-feature results by showing similar mechanisms that lead to an overall increase in tuning strength and decoder accuracy in neural populations. We have now added Figure R7 as Supplementary Figure 11 and included new text in the Results/Discussion sections highlighting the similarity of the results between iso and cross-feature adaptation (lines 379-383). As per reviewer's comment, we have corrected the issue regarding the X axis label.

Regarding reviewer's comment on the percentage values for the decoder accuracy, we calculated the change in decoder accuracy (DA) as a percentage, i.e., $[DA(\text{adapt}) - DA(\text{unadapt})] / DA(\text{unadapt})$ multiplied by 100. Hence, this measure can yield a range of values not necessarily restricted to values less than 50%. However, the reviewer would have been correct if we were reporting raw values, i.e., $[DA(\text{adapt}) - DA(\text{unadapt})]$. To avoid further confusion, we have relabeled the y axis in each of the plots to appropriately reflect the way the decoder accuracy was calculated. We thank the reviewer for helping us to bring clarity to our plots and analysis.

4. The discussion section is poor and needs to be more refined.

The authors have added a new Figure 5 to illustrate a possible mechanism for cross-feature adaptation. While I agree that this particular circuit explains cross-feature adaptation, it is inadequate given the generality of this circuit. For example, if the synapses of the V2 input encoding vertical gratings are stronger, we would expect color tuning to be weaker after adaptation. In this case, the sharpening of color tuning after cross-feature adaptation of the entire population would be negated. Overall, I don't think the discussion section is refined enough.

We appreciate reviewer's comment regarding the need to further refine the Discussion section. To address this concern, we have now added additional text in the Discussion section that addresses the importance, usefulness, and implications of cross-feature adaptation in local cortical circuits (see also our response to comment #1). Moreover, we have emphasized how our findings suggest new psychophysics and electrophysiological investigations to bridge the existing gap between cross-feature/multi-feature adaptation and perception during natural viewing.

Regarding reviewer's comment about the model in Figure 5, we would like to point out that Figure 5b presents a combination of a possible arrangement of feedforward inputs and adapting stimulus that explains our most striking finding, i.e., the emergence of tuning after adaptation to a largely dissimilar feature (Figure R3A). The reviewer is correct in stating that a set of inputs with a different distribution of synaptic strengths could lead to a loss of tuning as illustrated in Figure R3B. This is precisely what we observed in our analysis (Figure 3b). There is heterogeneity in the effects of cross-feature adaptation that gives rise to either tuning

Figure R3. (A) Schematic representation of adapting stimulus and input signal combination giving rise to emergence/sharpening of tuning. (B) Another example combination of input signals that combined with the adapting stimulus (vertical grating) leads to broadening of color tuning.

changes (i.e., emergence, sharpening, broadening, or disappearance of tuning) or leaving tuning characteristics unchanged. In fact, this heterogeneity likely reflects the underlying diversity in the structure of feedforward inputs to midlevel visual areas. Heterogeneity of neural effects is commonly observed in a wide range of electrophysiological studies including but not limited to attention where not all neurons exhibit a response increase. However, it is important to note that despite the diverse changes in neural tuning we report an overall significant increase in tuning strength and an improvement in stimulus discriminability through decorrelation of population responses. We have now included new text in the Discussion section that highlights the above points (lines 341-346).

Minor comments

13. Figure 4f. Which monkey is this data from? Why is the decoder performance for unadapted stimuli lower than zero?

I still don't understand why the decoder performance for unadapted stimuli was less than zero? I understand that the raw decoder performance is not less than zero. But why is decoder performance below chance level? This seems to be true for almost all experiments.

To address reviewer's comment regarding decoder accuracies being less than expected by chance (50%), we would first like to clarify how we ran the decoder analysis. For a pair of stimuli, we trained the decoder using the neural responses in 70% of the trials. The performance of the decoder was then tested on the remaining 30% of trials. We repeated this process 500 times, each time randomly selecting 70% of trials to train, and the rest of trials were used to test the decoder. This generates a distribution of decoder accuracy values for both unadapted and adapted conditions. The values in Figure 4d (main manuscript) denote the mean and s.e.m of this distribution. Figure R4A shows the distribution of decoder accuracy values (N = 500 iterations) using LDA classifier for both conditions in an example session. Note that the mean value of the distribution for unadapt condition is close to 50%. To check whether accuracy values less than 50% on certain iterations of the decoder were due to an artifact specific to our linear decoder, we implemented two additional non-linear decoders: Support Vector Machines (SVM) and K nearest neighbor (KNN) on the same example session used to generate Figure R4A. We observed that in both cases there were iterations that yielded decoder accuracies < 50% (Figure R4 B, C). Again, note that the mean values of decoder accuracies in the unadapt condition (blue) are very close to 50%. This demonstrates that less than chance accuracy values do not reflect an error or artifact in our analysis but they reflect instead an inherent property of classification algorithms. We believe that below chance values occur because the randomly sampled training set generates a sub-optimal model of the neural responses leading to higher than expected misclassifications (lower than expected accuracy) on the test data. However, the non-linear decoders show qualitatively similar increases in decoding performance post adaptation confirming that our findings of improvement in stimulus discriminability are general and hold for both linear and non-linear decoders.

Figure R4. (A) Probability density function of decoder accuracy values over multiple iterations (N = 500) for unadapt (blue) and adapt (red) conditions for an example session. Arrows represent the mean values of each distribution along with the corresponding percentage change in decoder accuracy (Δ). (B, C) Same as in A except for decoder accuracy was calculated using Support Vector Machine (SVM) and k-nearest neighbor (KNN) classifiers.

REVIEWERS' COMMENTS

Reviewer #1 (Remarks to the Author):

The authors have adequately responded to my remaining concerns. I find this study quite novel in its question and approach. Some of my questions point to further future studies in neurophysiology and psychophysics.

Reviewer #2 (Remarks to the Author):

The authors have adequately addressed all of my comments. I commend the authors for their diligent efforts. I have no further comments.